# Spatial redundancy transformer for self-supervised fluorescence image denoising

Xinyang Li [1,2,3,9], Xiaowan Hu[2,9], Xingye Chen[1,3,4,9], Jiaqi Fan[2,5], Zhifeng Zhao[1,3], Jiamin Wu [1,3,6,7] ✉, Haoqian Wang [2,8] ✉ & Qionghai Dai [1,3,6,7] ✉

Fluorescence imaging with high signal-to-noise ratios has become the foundation of accurate visualization and analysis of biological phenomena. However, the inevitable noise poses a formidable challenge to imaging sensitivity. Here we provide the spatial redundancy denoising transformer (SRDTrans) to remove noise from fluorescence images in a self-supervised manner. First, a sampling strategy based on spatial redundancy is proposed to extract adjacent orthogonal training pairs, which eliminates the dependence on high imaging speed. Second, we designed a lightweight spatiotemporal transformer architecture to capture long-range dependencies and high-resolution features at low computational cost. SRDTrans can restore high-frequency information without producing oversmoothed structures and distorted fluorescence traces. Finally, we demonstrate the state-of-the-art denoising performance of SRDTrans on single-molecule localization microscopy and two-photon volumetric calcium imaging. SRDTrans does not contain any assumptions about the imaging process and the sample, thus can be easily extended to various imaging modalities and biological applications.

The rapid development of intravital imaging techniques enables researchers to observe biological structures and activities at micrometer and even nanometer scales[1,2]. As an imaging method with great prevalence, fluorescence microscopy has contributed to the discovery of a series of new physiological and pathological mechanisms due to its high spatiotemporal resolution and molecular specificity[3–5]. The fundamental goal of fluorescence microscopy is to obtain clean and sharp images containing sufficient information about the sample, which can guarantee the accuracy of downstream analysis and support convincing conclusions. However, limited by multiple biophysical and biochemical factors (for example, labeling concentration, fluorophore brightness, phototoxicity, photobleaching and so on), fluorescence imaging is conducted in photon-limited conditions and the inherent photon shot noise severely degrades the image

signal-to-noise ratio (SNR), especially in low-illumination and high-speed observations[6].

Various methods have been proposed to remove noise from fluorescence images. Conventional denoising algorithms based on numerical filtering and mathematical optimization suffer from unsatisfactory performance and limited applicability[7,8]. In the past few years, deep learning has shown remarkable performance in image denoising[9,10]. After iterative training on a dataset with ground truth (GT), deep neural networks can learn the mapping between noisy images and their clean counterparts. Such a supervised manner depends heavily on paired GT images[11–15]. When observing the activity of living organisms, obtaining pixel-wise registered clean images is a great challenge because the sample often undergoes fast dynamics. To alleviate this contradiction, some self-supervised methods have been proposed

[1]Department of Automation, Tsinghua University, Beijing, China. [2]Tsinghua Shenzhen International Graduate School, Tsinghua University, Shenzhen, China. [3]Institute for Brain and Cognitive Sciences, Tsinghua University, Beijing, China. [4]Research Institute for Frontier Science, Beihang University, Beijing, China. [5]Department of Electronic Engineering, Tsinghua University, Beijing, China. [6]Beijing Key Laboratory of Multi-dimension and Multi-scale Computational Photography (MMCP), Tsinghua University, Beijing, China. [7]IDG/McGovern Institute for Brain Research, Tsinghua University, Beijing, China. [8]The Shenzhen Institute of Future Media Technology, Shenzhen, China. [9]These authors contributed equally: Xinyang Li, Xiaowan Hu, Xingye Chen. ✉e-mail: wujiamin@tsinghua.edu.cn; wanghaoqian@tsinghua.edu.cn; qhdai@tsinghua.edu.cn

for more applicable and practical denoising in fluorescence imaging[6,16–23]. Among them, the first kind of methods rely on the similarity between adjacent frames[6,16–18]. But when the sample changes very quickly or the imaging speed is too slow, the time-lapse data cannot provide enough temporal redundancy. This is a common problem in volumetric imaging as the volume rate decreases proportionally to the number of imaging planes. The dissimilarity between adjacent frames will result in inferior performance and distorted structures and fluorescence kinetics. The other kind of methods learn to denoise only from spatially adjacent pixels in two-dimensional frames[19–23]. However, without utilizing endogenous temporal correlations, these methods perform poorly on time-lapse imaging. Therefore, to achieve better denoising performance, the ability to simultaneously extract global spatial information and long-range temporal correlations is essential, which is lacking in convolutional neural networks (CNNs) because of the locality of convolutional kernels[24]. Moreover, the inherent spectral bias makes CNNs tend to fit low-frequency features preferentially while ignoring high-frequency features, inevitably producing oversmoothed denoising results[25].

Here we present the spatial redundancy denoising transformer (SRDTrans) to address these dilemmas. On the one hand, a spatial redundancy sampling strategy is proposed to extract three-dimensional (3D) training pairs from the original time-lapse data in two orthogonal directions. This scheme has no dependence on the similarity between two adjacent frames, so SRDTrans is applicable to very fast activities and extremely low imaging speed, which is complementary to our previously proposed DeepCAD that leverages temporal redundancy[6,18]. On the other hand, we designed a lightweight spatiotemporal transformer network to fully exploit long-range correlations. The optimized feature interaction mechanism allows our model to obtain high-resolution features with a small number of parameters. Compared with classical CNNs, the proposed SRDTrans has stronger abilities for global perception and high-frequency maintenance, enabling the revelation of fine-grained spatiotemporal patterns that were previously indiscernible. We demonstrate the superior denoising performance of SRDTrans on two representative applications. The first one is single-molecule localization microscopy (SMLM) with adjacent frames being random subsets of fluorophores[26]. The other one is two-photon calcium imaging of large 3D neuronal populations with a volumetric speed as low as 0.3 Hz. Extensive qualitative and quantitative results indicate that SRDTrans can serve as a fundamental denoising tool for fluorescence imaging to observe various cellular and subcellular phenomena.

## Results

### Principle of SRDTrans

The self-supervised framework of SRDTrans is shown schematically in Fig. 1a. For spatial redundancy sampling, spatially adjacent substacks are sampled by orthogonal masks from the original low-SNR image stack. Each target is adjacent to the input stack horizontally or vertically to exploit spatial correlations fully and isotropically. A simplified implementation of sampling is depicted in Fig. 1b. As the noise of adjacent pixels is independent while the signals are closely correlated, the substack filled by the central pixels can be used as the training input and the other two spatially adjacent substacks are used as corresponding targets to optimize the network parameters. Compared with other methods[19,20,22], our sampling strategy is more effective and comprehensive in preserving both spatial and temporal information (Supplementary Figs. 1 and 2). In the inference stage, low-SNR stacks will be fed into pre-trained SRDTrans models without spatial downsampling. To overcome the locality of convolutional kernels, we constructed a transformer network to capture endogenous non-local spatial features and long-range temporal dependencies using the self-attention mechanism (Fig. 1c). The restoration of each pixel can simultaneously integrate the temporal information of all frames and the spatial information of all pixels, even if they are far from each other. Besides, the network does not contain any spatial downsampling module, allowing more high-frequency components to flow through the network and avoiding the loss of spatial resolution. Furthermore, as the amount of data in fluorescence imaging is often very large, sometimes at the petabyte scale, the transformer network was designed to be as lightweight as possible to relieve the computational burden of large-scale data processing. Compared with other transformer networks[27–30], our architecture can achieve the best performance with more than one order of magnitude fewer parameters (Supplementary Tables 1 and 2). The lightweight architecture of SRDTrans also makes it easy to train a good model even with a small amount of training data (for example, 500 frames, 490 × 490 pixels each frame), relieving the pressure of capturing large-scale datasets (Supplementary Fig. 3).

To demonstrate the predominance of our transformer network over CNNs, we generated simulated calcium imaging data (Methods and Supplementary Fig. 4) and used them to train a 3D U-Net[31], as well as the transformer of SRDTrans, using the same spatial redundancy sampling strategy. We term the former as spatial redundancy denoising CNN (SRDCNN). The visualized feature maps of deep layers intuitively show the superiority of SRDTrans in revealing fine-grained patterns (Fig. 1d). As features flow through the network, the limited receptive field of convolutional kernels makes CNN-based methods focus on only rough features while our transformer architecture still has a strong perception of sophisticated structures. We also compare the denoised images of the two architectures (Fig. 1e). The result of SRDCNN is obviously oversmoothed, especially in regions with sharp edges, which is a manifestation of spectral bias that the low-frequency information is overfitted while the high-frequency information is hardly preserved (Supplementary Fig. 5). This deficiency is largely alleviated by SRDTrans, and more subcellular structures such as dendritic fibers can be restored accurately. The intensity profile deconstructed from the SRDTrans denoised image is more consistent with the GT (Fig. 1f). Moreover, lacking the ability to capture long-range temporal

**Fig. 1 | Principle of SRDTrans and performance evaluation. a**, Self-supervised training strategy of SRDTrans. The original low-SNR stack of $H \times W \times T$ pixels is sampled by orthogonal masks, producing three downsampled substacks (input, target 1 and target 2) of $H/2 \times W/2 \times T$ pixels. The 'input' substack is fed into the transformer network, and the corresponding output is compared with the 'target' substacks to calculate the loss function for parameter optimization. **b**, Simplified schematic of spatial redundancy sampling ($H = 4$, $W = 4$, $T = 1$). A 4 × 4 patch is split into four 2 × 2 blocks and three adjacent pixels are randomly selected in each block. The central pixel (labeled as '2') is horizontally or vertically adjacent to the other two pixels (labeled as '1' and '3'). **c**, The architecture of the lightweight spatiotemporal transformer. It consists of a temporal encoder module, an STB and a temporal decoder module. Each temporal encoder compresses the temporal scale ($t$) of the input by a factor of $r$ ($r = 4$ in this work) using convolution. In the STB module, the input is divided into small patches, and different feature maps of the same spatial position are stitched together in the patch flattening layer. The position embedding layer records the spatial position of each patch so that it can be mapped back after the global interaction in the multi-head self-attention layer. The self-attention mechanism can calculate the spatiotemporal correlation between all local patches. The output of the STB module will be uncompressed to the original temporal scale by the following temporal decoder module. **d**, Visualizing the feature responses in SRDCNN (the last layer of STB) and SRDTrans (the last layer of 3D U-Net). SRDCNN represents the method that replaces the transformer network in SRDTrans with a 3D U-Net. Scale bar, 60 μm. **e**, Comparing the denoising performance of SRDCNN and SRDTrans on simulated calcium imaging data (30 Hz). Scale bars, 40 μm for the whole FOV and 10 μm for magnified views. **f**, Pixel intensity along the red dashed line in **e**. **g**, Evaluating the ability of SRDCNN and SRDTrans to capture long-range dependencies. Models were trained and validated on simulated calcium imaging data (30 Hz) of different input temporal scales ($T$). All values are shown as mean ± s.d. ($N = 6,000$ independent frames).

correlations also drags down the denoising performance of SRDCNN on time-lapse imaging data. For SRDTrans, the output SNR continuously grows as the input temporal scale ($T$) increases (Fig. 1g). A more comprehensive evaluation of the influence of input temporal scale indicates that SRDTrans can make full use of the information offered by temporally distant pixels (Supplementary Figs. 6 and 7). We also investigated the generalization ability of the proposed method by cross-dataset and cross-modality validation, which shows that training on data with the same SNR and imaging modality can obtain the best denoising performance (Supplementary Fig. 8). To verify the practicality of SRDTrans at extremely low imaging speed, we compared the

performance of methods combining different networks and sampling schemes on simulated calcium imaging data sampled at 0.3 Hz (Supplementary Video 1). When the similarity between adjacent frames is low, using spatial redundancy sampling is more reasonable (Supplementary Table 3). Succinctly, the synergy between spatial redundancy sampling and dedicated transformer architecture endows SRDTrans with the ability to restore high-resolution structures and fast dynamics.

## High-performance SMLM with SRDTrans denoising

Given $N$ detected fluorescence photons, the lower bound of the precision of SMLM scales to $1/\sqrt{N}$ (ref. 26), which is the mathematical

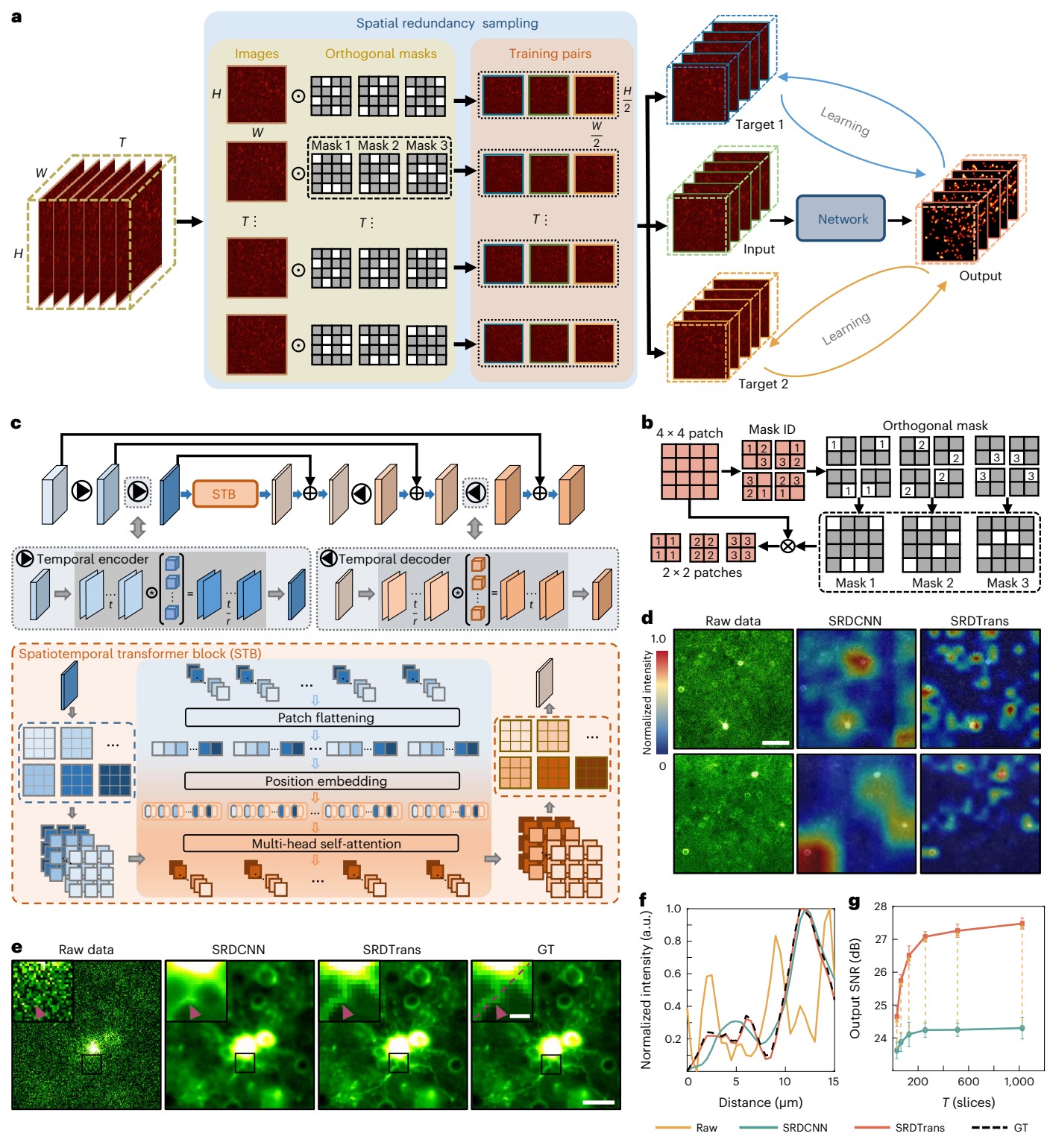

formula of the shot-noise limit or the well-known standard quantum limit[32,33]. This indicates that the fundamental physical limit of localization precision is shot noise. To investigate the benefits that our denoising method can bring to SMLM, we first applied SRDTrans to simulated stochastic optical reconstruction microscopy (STORM) data with GT for quantitative evaluation[34,35]. The noise-free single-molecule-emission images were synthesized by TestSTORM[36] and corresponding noisy images with different SNRs were then generated by applying different levels of mixed Poisson–Gaussian noise (Methods and Supplementary Fig. 9). For the image stack of each SNR, we trained a specific model for it and then processed it using the model to obtain the denoised image stack. Quantitative comparisons of the visualized images (Fig. 2a and Supplementary Video 2) and the extracted intensity profiles (Fig. 2b) show that the results of SRDTrans are highly consistent with the GT. Over a wide range of imaging SNRs, including some extremely low-SNR conditions, SRDTrans can substantially improve the image quality evaluated by the SNR at the pixel-intensity level and structural similarity (SSIM) at the perception level (Fig. 2c). Compared with other self-supervised methods[16–22], SRDTrans can better preserve the distribution and intensity of emitters owing to its strong ability in exploiting high-resolution features and long-range dependencies (Supplementary Fig. 10).

Next, we evaluate the improvement of single-molecule localization performance with the enhancement of the image SNR. To reconstruct super-resolved images, we applied ThunderSTORM[37], one of the most frequently used localization software with an excellent balance between accuracy and execution runtime[38,39]. The image reconstructed from the original noisy data is contaminated by noise and contains many misidentified molecules (Fig. 2d). By contrast, the reconstructed image from SRDTrans denoised images reveals clear and continuous cytoskeletal filaments that are not previously recognizable because of suppressed localization error and improved resolution (Fig. 2e and Supplementary Fig. 11). For better quantitative analysis, we matched the detected fluorescent molecules with the GT using the Hungarian algorithm[39]. From raw images, few fluorescent molecules can be detected and most of them are wrongly localized. After SRDTrans denoising, almost all molecules can be detected in high agreement with the GT (Fig. 2f and Supplementary Fig. 12). Using the Jaccard index and root-mean-squared error (r.m.s.e.) as the metrics to quantify the proportion of correctly detected molecules and the localization accuracy of those detected molecules, respectively, we found that the Jaccard index was improved by ~6-fold ($85.7 \pm 3.51\%$ versus $14.1 \pm 2.76\%$, mean ± s.d.) and r.m.s.e. was reduced by ~3.4-fold ($24.86 \pm 3.24$ nm versus $85.94 \pm 7.76$ nm) after denoising (Fig. 2g). From a more comprehensive perspective, we further adopted the metric termed efficiency that combines Jaccard index and r.m.s.e.[39]. The results show that SRDTrans improved the efficiency of single-molecule localization from $-21.54 \pm 5.71$ to $71.33 \pm 3.38$ (Fig. 2h), fully demonstrating the benefits of SRDTrans on SMLM.

We further applied SRDTrans to experimental SMLM data of microtubules to validate its ability in revealing subcellular structures. Raw frames were captured with low excitation power and short exposure time to reduce phototoxicity and emitter density. The experimentally obtained single-molecule-emission images and SRDTrans denoised images are shown in Fig. 3a. Disturbed by the noise, the reconstruction algorithm can hardly localize the fluorescent molecules in the raw frames. The reconstructed super-resolution image contains many erroneous spots and cannot reveal any meaningful structures (Fig. 3b). In comparison, SRDTrans can effectively suppress the noise and remove localization artifacts in the reconstructed image, making the distribution and extension directions of microtubules visible. We computed the Fourier-ring correlation (FRC)[40,41] curve to quantify the resolution from the SMLM images. The image resolution is defined as the inverse of the spatial frequency at the intersection of the FRC curve and the threshold line (~0.143). Benefitting from the removal of noise, the resolution of SRDTrans denoised data was improved from 52.4 nm to 36.0 nm (Fig. 3c) and the localization uncertainty was reduced from $8.0 \pm 6.88$ nm to $5.0 \pm 1.34$ nm (Fig. 3d). In addition to the data acquired by our instrument, we also used SRDTrans to denoise publicly available SMLM data contributed by other laboratories[42] (Fig. 3e). The reconstructed super-resolution images indicate that SRDTrans can eliminate the artifacts and bring more complete organelle structures (Fig. 3f,g). We applied Gaussian fitting to the intensity profile perpendicular to the microtube filaments and measured the full-width at half-maximum (FWHM) to quantify the image resolution (Fig. 3h). The SRDTrans denoised data show improved resolution as the averaged FWHM dropped from $187.89 \pm 22.22$ nm to $60.96 \pm 7.51$ nm (Fig. 3i). As SMLM is heading towards live-cell imaging and long-term observation[43] our denoising method promises to be a beneficial tool to reduce the laser power by resolving fluorescent molecules from very-low-SNR frames. For 3D SMLM, as the axial positions of molecules are estimated through point-spread-function engineering[26], SRDTrans is expected to offer great help by resolving single-molecule-emission patterns from low-SNR images.

## Applying SRDTrans to two-photon volumetric calcium imaging

In multiphoton imaging, the volumetric imaging speed decreases linearly as the number of scanning planes increases. Thus, the achievable sampling rate for observing neuronal populations with large axial ranges is often quite low, making the denoising methods that rely on the similarity between temporally adjacent frames infeasible[6,16–18]. However, SRDTrans provides an opportunity to restore the highly degraded fluorescence signals in large-scale volumetric calcium imaging by utilizing the similarity between spatially adjacent pixels. To evaluate the denoising performance of SRDTrans on calcium imaging with different sampling rates, we generated realistic calcium imaging data with synchronized GT using neural anatomy and optical microscopy (NAOMi)[44]. We started from denoising high-sampling-rate (30 Hz) data and found that SRDTrans can effectively remove noise and recover previously indiscernible structures such as soma, neurites and vascular shadows (Fig. 4a, Supplementary Fig. 13 and Supplementary Video 3). The enhancement is manifested not only in the visual effect but also, more importantly, in the accurate restoration of pixel intensities

**Fig. 2 | Validation of SRDTrans on simulated SMLM data. a**, Single-molecule-emission images before and after denoising. Left: raw data. Middle: SRDTrans denoised data. Right: GT. Magnified views of boxed regions show the emission pattern of a bunch of fluorescent molecules. Scale bars, 2 µm for the whole FOV and 0.5 µm for magnified views. The SNR value of the raw data and denoised data are noted. **b**, Intensity profiles along the white dashed lines in **a**. **c**, Quantitative evaluation of the denoising performance with SNR and SSIM. Left: image SNR before and after denoising. Right: image SSIM before and after denoising. Each data point shows the statistical result of 24,000 frames. All values are shown as mean ± s.d. ($N = 24{,}000$ independent frames). **d**, Reconstructed super-resolution images of microtubules before and after denoising. Left: the image reconstructed from raw data. Middle: the image reconstructed from SRDTrans denoised data. Right: GT. Scale bar, 5 µm. **e**, Merged image of the yellow boxed region in **d**. Magenta, the image reconstructed from raw data; green, the image reconstructed from SRDTrans denoised data; red, GT. The overlapping positions of red and green appear yellow. Scale bar, 1 µm. **f**, Consistency analysis of the localized fluorescent molecules in raw images (left) and SRDTrans denoised images (right) relative to the GT. A magnified view of the boxed region is shown at the bottom left of each panel. **g**, Tukey box-and-whisker plots (Methods) showing the localization precision quantified with the Jaccard index (left, higher is better) and r.m.s.e. (right, lower is better) before and after SRDTrans denoising ($N = 5{,}000$ independent molecules). **h**, Evaluating the performance of single-molecule localization before (blue) and after (orange) denoising with a more comprehensive metric termed efficiency.

(Fig. 4b). Visualization in the frequency domain (calculated by discrete Fourier transform) shows that SRDTrans can restore most of the frequency components (Fig. 4c), especially the high-frequency components lost by DeepCAD[6,18] and DeepInterpolation[17], thereby leading to high performance in denoising calcium imaging data (Supplementary Fig. 14). Such a remarkable denoising capability can be maintained over a wide range of input SNRs (from −2.08 dB to 17.68 dB), and the average SNR improvement is about 22 ± 2.47 dB (Fig. 4d). We also verified the

performance of SRDTrans on experimentally obtained calcium imaging data with a synchronized high-SNR (tenfold photons) reference[6], which shows that the neuronal structures and dynamics swamped by noise can be restored authentically (Supplementary Fig. 15).

Then we investigate the denoising performance of SRDTrans on calcium imaging data sampled at 0.3 Hz, which is 100 times lower than the imaging speed demonstrated above. Bilateral assessments in both the space domain and the frequency domain reveal that SRDTrans can

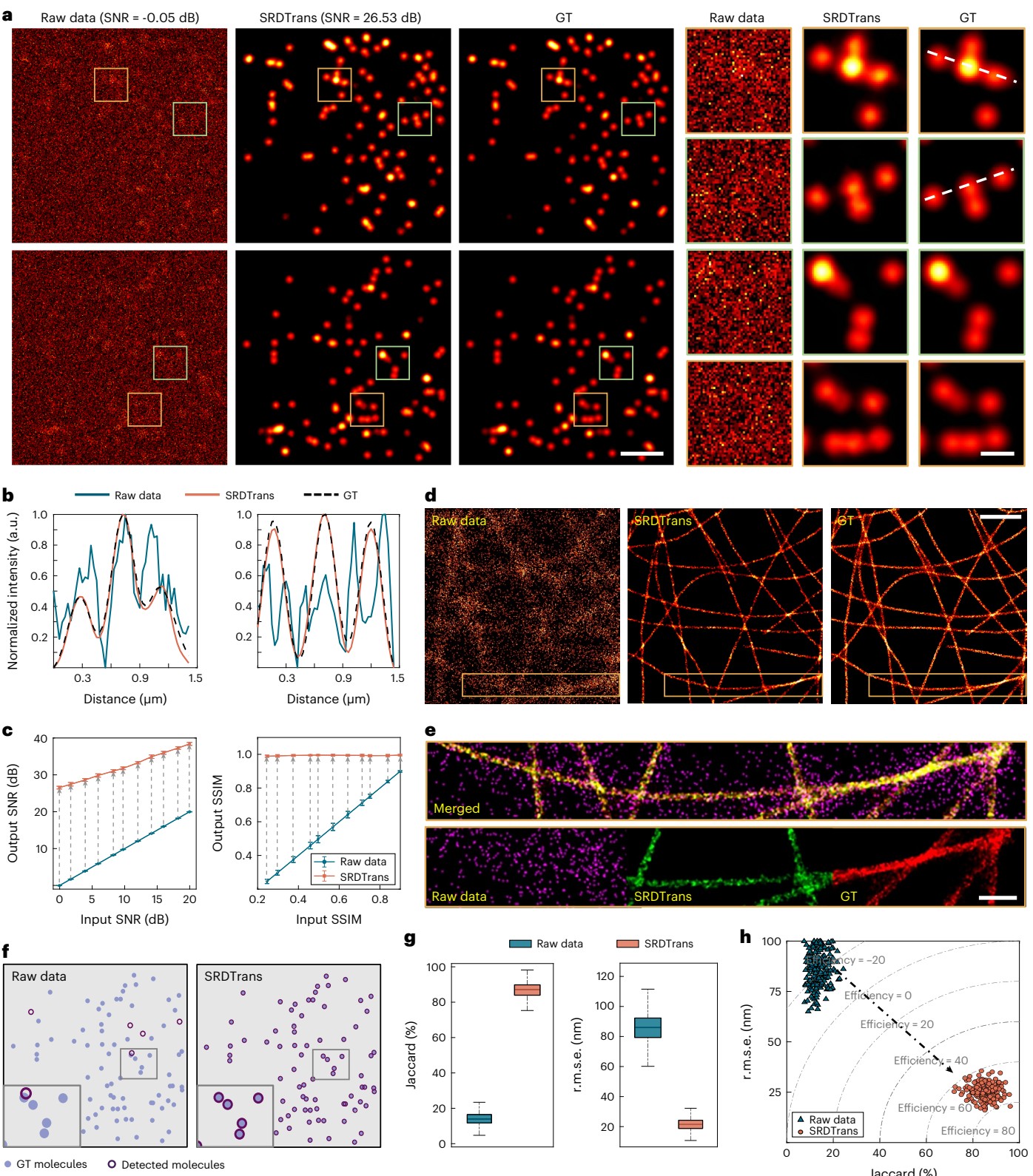

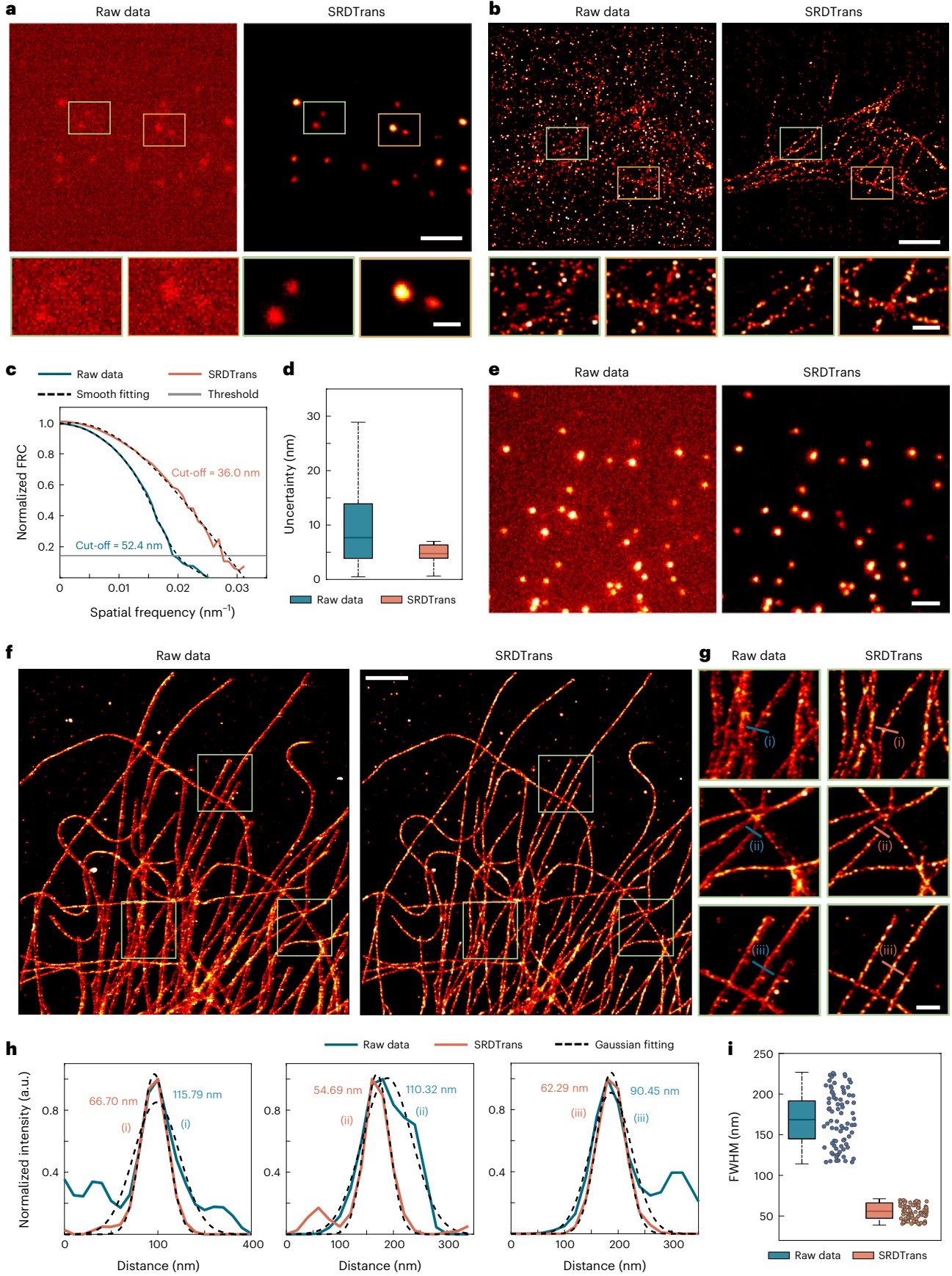

**Fig. 3 | Applying SRDTrans to experimental SMLM data. a**, Experimentally obtained single-molecule-emission images. Left: raw data. Right: SRDTrans denoised data. The magnified views of two boxed regions are shown below each image. Scale bars, 2 µm for the whole FOV and 0.5 µm for magnified views. **b**, Reconstructed super-resolution images. The microtubules in fixed BSC-1 cells were labeled with Cy5. Scale bars, 2 µm for the whole FOV and 0.5 µm for magnified views. **c**, FRC curves of the raw reconstructed image (blue) and the SRDTrans enhanced reconstructed image (orange). The estimated resolution (52.4 nm for raw image and 36.0 nm for SRDTrans denoised image) is the inverse of the spatial frequency where the FRC curve drops below the cut-off threshold (-0.143). **d**, Tukey box-and-whisker plots (Methods) showing the localization uncertainty before and after denoising ($N = 1,048,575$ detected molecules for

raw data, $N = 395,908$ detected molecules for SRDTrans). The uncertainty was calculated by the ThunderSTORM plugin. **e**, Single-molecule-emission images from the open-source platform ShareLoc[51]. Left: raw data. Right: SRDTrans denoised data. Scale bar, 2 µm. **f**, Reconstructed super-resolution images of microtubules (immuno-labeled with Alexa 647). Scale bar, 2 µm. **g**, Magnified views of boxed regions. Scale bar, 0.5 µm. **h**, Intensity profiles perpendicular to the microtubule filaments indicated in **g**. Blue line, raw data; orange lines, SRDTrans denoised data; dashed line, the Gaussian fitting result. The corresponding FWHM value is quantified as $2.335\sigma$, where $\sigma$ denotes the standard deviation of the Gaussian fitting result. **i**, Tukey box-and-whisker plots (Methods) showing the FWHM of randomly selected filaments (blue, raw data; orange, SRDTrans denoised data; $N = 80$ independent filament segments).

accurately retrieve the fluorescence signals from the original highly degraded images without structural blurring and frequency deficiency (Fig. 4e and Supplementary Fig. 14). When the sampling rate is much lower than the fluorescence dynamics, the large discrepancy between the signals in two adjacent frames cannot provide the temporal correlation required by DeepCAD and DeepInterpolation, so they are not accurate enough to be used in conditions of low imaging speed or fast activity (Supplementary Fig. 16). To figure out how SRDTrans works at different imaging speeds, we performed an ablation study on different sampling strategies and network architectures (Fig. 4f and Supplementary Table 3). The results indicate that spatial redundancy sampling performs better at low imaging speeds, whereas temporal redundancy sampling performs better at high imaging speeds. Almost equally for all imaging speeds, our transformer architecture offers an additional improvement (~$2.05 \pm 0.27$ dB) over conventional CNNs. In general, the synergistic combination of the spatial redundancy sampling and the transformer architecture in SRDTrans provides better performance than DeepCAD at all imaging speeds (Supplementary Figs. 17 and 18). In the time domain, the superior ability of SRDTrans can reveal high-fidelity calcium transitions without distorting fluorescence kinetics (Fig. 4g). Moreover, we also simulated fast-moving objects to imitate migrating cells that are widely existed in living organisms. The quantitative evaluation shows that SRDTrans can preserve the structure of densely distributed objects even if they are moving at a speed of up to 9 pixels per frame (Supplementary Fig. 19 and Supplementary Video 4), alleviating the shortage of denoising methods for fast-moving cells and organelles.

Finally, we went a step further in denoising calcium imaging data by applying SRDTrans to volumetric recordings, which is not achievable for other self-supervised denoising methods[6,17,18] because of their heavy reliance on high sampling rates. We used transgenic mice expressing the GCaMP6f genetically encoded calcium indicator[45] and imaged a brain volume of $500 \times 500 \times 200$ µm³ in the mouse cortex using a two-photon microscope. We scanned 100 planes with a frame rate of 30 Hz, and thus the overall volume rate was 0.3 Hz. For the denoising of volumetric calcium imaging data, we extracted all the frames of each imaging plane and reorganized them into a separate time-lapse

($xy$–$t$) stack. The time-lapse stacks of all imaging planes were used for network training. A 3D visualization of the neural volume shows that the spatial profiles and firing states of the neurons can be revealed after denoising, which otherwise would be swamped by severe shot noise (Fig. 5a and Supplementary Video 5). For better comparison, we present the snapshots of a certain imaging plane at two different moments. With the enhancement of SRDTrans, the structure and distribution of the neurons become clearly observable (Fig. 5b). We also extracted the fluorescence traces along the time axis and found that a large number of calcium transients can be restored after denoising (Fig. 5c). The dramatically improved SNR would propel the decoding of underlying neural activity from fluorescence signals. As neural circuits in the mammalian brain are spatially coordinated and temporally orchestrated, deciphering the function of large neuronal ensembles requires large-scale volumetric imaging with a high SNR. The superior denoising performance of SRDTrans provides an opportunity to implement high-sensitivity volumetric calcium imaging for investigating functionally concerted neurons and recognizing circuit motifs, especially those distributed across multiple cortical layers.

## Discussion

SRDTrans does not rely on any assumptions about the contrast mechanism, noise model, sample dynamics and imaging speed. Thus, it can be readily extended to other biological samples and imaging modalities (Supplementary Fig. 20), such as membrane voltage imaging, single-protein detection, light-sheet microscopy, confocal microscopy, light-field microscopy and super-resolution microscopy[46–51]. The limitation of SRDTrans lies in the basic assumption that neighboring pixels should have approximate structures. If the spatial sampling rate is too low to provide enough redundancy, SRDTrans would fail. Another potential risk is the generalization ability as the lightweight network architecture of SRDTrans is more suitable for specific tasks. We believe training specific models for specific data is the most reliable way to use deep learning for fluorescence image denoising. Therefore, a new model should be trained to ensure optimal results when the imaging parameter, modality and sample change.

**Fig. 4 | Evaluating the performance of SRDTrans on simulated calcium imaging data. a**, SRDTrans denoised calcium imaging data sampled at 30 Hz. Magnified views show the neural activity of the yellow boxed region in a short period (~2 s). Left: the original low-SNR data. Middle: SRDTrans denoised data. Right: GT. Scale bars, 60 µm for the whole FOV and 30 µm for magnified views. The magenta arrowhead indicates a dendritic fiber and the yellow arrowhead indicates two neighboring somas. **b**, Pixel intensity along the yellow dashed line in **a**. Top left: raw data. Middle left: SRDTrans denoised data. Bottom left: GT. Right: plotting the three intensity profiles in one coordinate. The similarity with GT is quantified by Pearson correlation coefficients ($R$). **c**, Frequency spectrum calculated by discrete Fourier transform before and after denoising. Magnified views of the boxed regions show the frequency components within the optical transfer function. The similarity in the frequency domain is quantified by LFD. **d**, The performance of SRDTrans at different SNR levels. All values are shown as mean ± s.d. ($N = 6,000$ independent frames). **e**, Comparing the performance of

DeepCAD and SRDTrans on calcium imaging data sampled at 0.3 Hz. Magnified views show the neural activity of yellow boxed regions in a 20 s time window. Scale bars, 100 µm for the whole FOV and 40 µm for magnified views. The yellow and purple arrowheads point to a firing neuron and a resting neuron, respectively. **f**, Ablation experiments to investigate the effects of different sampling strategies and network architectures. SRDTrans (orange) uses spatial redundancy sampling and a lightweight spatiotemporal transformer. DeepCAD (purple) combines temporal redundancy sampling and a CNN (3D U-Net). SRDCNN (green) is the method combining spatial redundancy sampling and a CNN (3D U-Net). All values are shown as mean ± s.d. ($N = 1,000$ independent frames for each frame rate). **g**, Fluorescence traces ($F$) extracted from 50 randomly selected neuronal pixels. The similarity with GT is quantified by Pearson correlation coefficients ($R$). Top: traces extracted from raw data. Middle: traces extracted from DeepCAD denoised data. Middle bottom: traces extracted from SRDTrans denoised data. Bottom: GT.

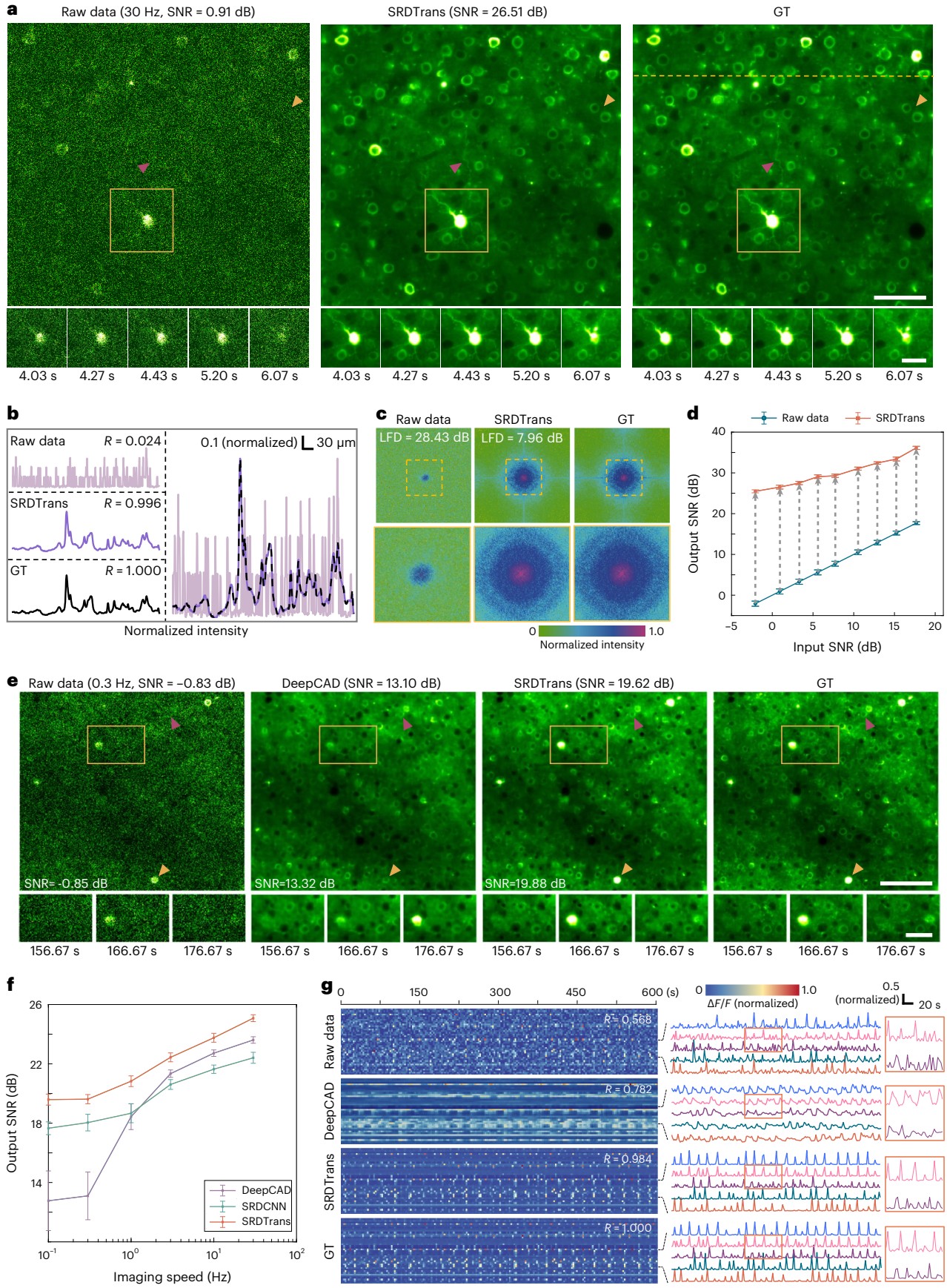

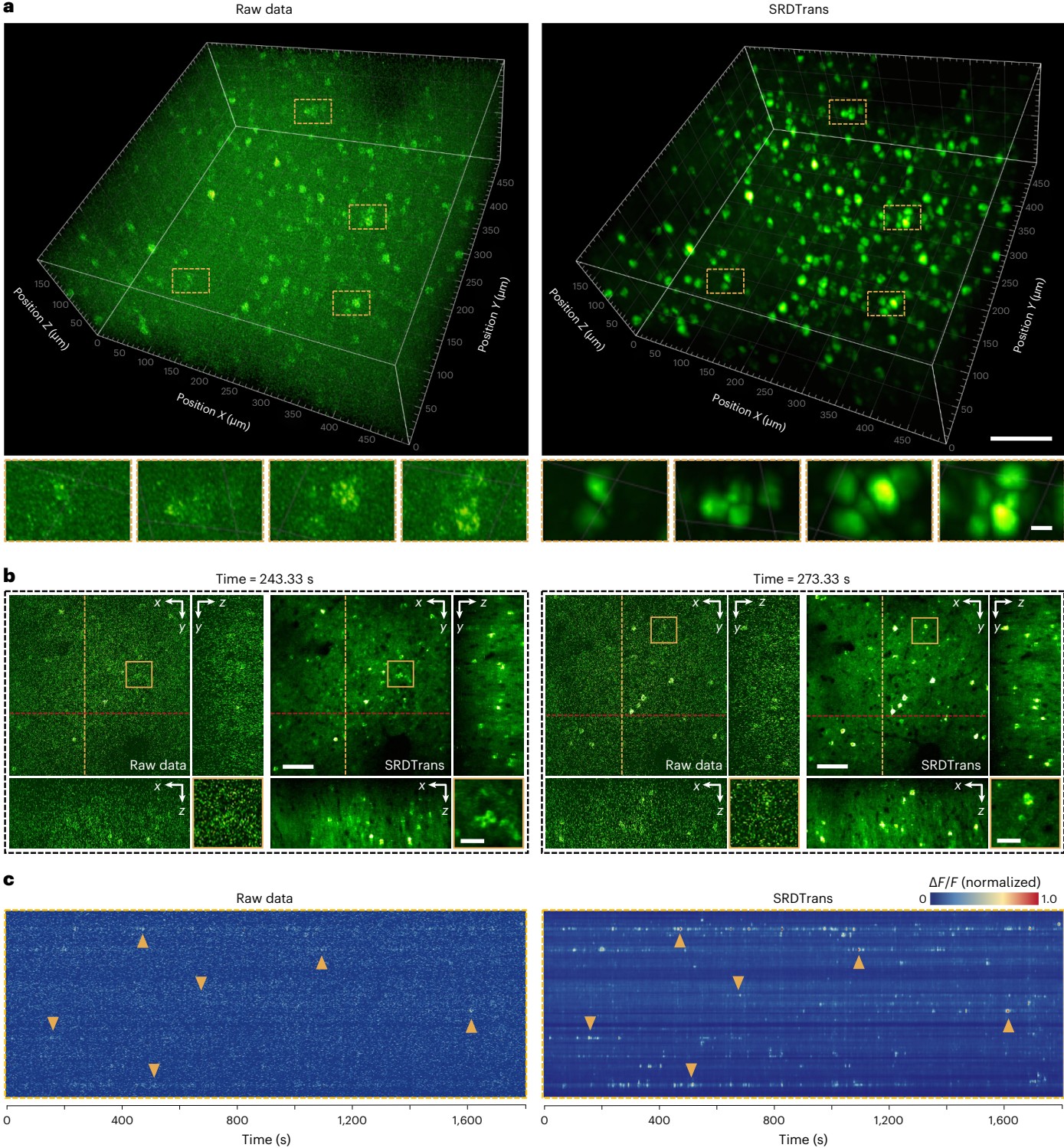

**Fig. 5 | High-sensitivity calcium imaging of large neural volumes. a**, Three-dimensional visualization of the neural activity of a 510 × 510 × 200 μm³ volume (100 planes, 0.3 Hz volume rate) in the mouse cortex. Left: the original low-SNR volume. Right: the same volume denoised with SRDTrans. Magnified views of yellow boxed regions are shown under each snapshot. Scale bars, 100 μm for the whole FOV and 10 μm for magnified views. **b**, Raw frames and SRDTrans denoised counterparts of a single imaging plane at two different moments. The *x*–*z* and *y*–*z* cross-sections of the volume are shown alongside each *x*–*y* plane. Magnified views of yellow boxed regions are shown at the bottom right of the images. Scale bars, 70 μm for the whole FOV and 20 μm for magnified views. **c**, Fluorescence traces (*F*) extracted from all pixels on the yellow dashed line in **b**. Left: traces extracted from raw data. Right: traces extracted from SRDTrans denoised data. Yellow arrowheads point to some representative calcium transients.

As the development of fluorescence indicators heads towards faster kinetics to monitor biological dynamics at the millisecond scale[52,53], the imaging speed required to record these fast activities is continuously growing. Obtaining adequate sampling rates is becoming more and more challenging for denoising methods relying on temporal redundancy. Our rationale is to fill the niche by seeking to utilize spatial

redundancy as an alternative to enable self-supervised denoising in more imaging applications. Although the perfect case for spatial redundancy sampling is that the spatial sampling rate is two times higher than the Nyquist sampling of the diffraction limit to ensure that two adjacent pixels have nearly identical optical signals, the endogenous similarity between two spatially downsampled substacks is sufficient to guide the training of the network in most cases. However, this does not mean that the proposed spatial redundancy sampling strategy can fully replace temporal redundancy sampling, as the ablation study (Fig. 4f) shows that, if equipped with the same network architecture, the temporal redundancy sampling can achieve better performance in high-speed imaging. The superiority of SRDTrans over DeepCAD at high imaging speeds is actually attributed to the transformer architecture. In general, spatial redundancy and temporal redundancy are two complementary sampling strategies to enable self-supervised training of denoising networks for fluorescence time-lapse imaging. Which sampling strategy is used depends on which kind of redundancy is more sufficient in the data. It is noteworthy that there are still many cases where neither redundancy is sufficient to support current sampling strategies. Developing specific or more general self-supervised denoising methods is of persistent value for fluorescence imaging.

## Methods

### The spatial redundancy sampling strategy

In SRDTrans, we employed a spatial redundancy sampling strategy to produce training pairs. The detailed implementation for generating training pairs in SRDTrans is shown in Fig. 1b and Supplementary Fig. 1d. For each image inside an input training stack with $H \times W \times T$ pixels ($H$, $W$ and $T$ are the height, width and length of the input image stack), we spatially divided it into many adjacent small patches with $2 \times 2$ pixels. Next, we randomly selected three adjacent pixels from each patch. The central pixel was used to compose the input substack and the two spatially adjacent pixels were used to compose the two target substacks. After traversing all patches, we can finally obtain three downsampled substacks with the size of $H/2 \times W/2 \times T$ pixels. As the fluorescence signals of spatially adjacent pixels are closely correlated, the input substack and each target substack can be considered as two independent samples of the same underlying pattern. Thus, the input substack and the two target substacks can form two training pairs, which can be used for the self-supervised training of denoising networks (Supplementary Note 1).

### Network architecture and loss function

The transformer architecture of SRDTrans is composed of three parts: a temporal encoder module, a spatiotemporal transformer block (STB) and a temporal decoder module (Fig. 1c). The temporal encoder module is equipped with two temporal encoders. Each temporal encoder can compress the temporal scale of the input substack by a factor of $r$ ($r$ is 4 in this work) using a convolutional layer with $3 \times 3$ kernels. In contrast to the U-Net-type architectures with many upsampling and downsampling operations, SRDTrans does not reduce the size of feature maps in these encoders (Supplementary Fig. 21). Thus, for an input substack with a size of $H/2 \times W/2 \times T$ pixels, the output size of the temporal encoder module is $H/2 \times W/2 \times T/r^2$. The output from the temporal encoder module will be fed into the STB to extract global information both in space and time. The STB contains a temporal transformer block and a spatial transformer block (Supplementary Fig. 22). Specifically, in the temporal transformer block, the input is divided into patches with the size of $p \times p \times T/r^2$ ($p$ is 7 in this work). These patches are first flattened into one-dimensional vectors and input into the position embedding layer, where spatial concatenation and linear transformation are performed. Two multi-head self-attention layers are then cascaded to extract temporal correlations inside the data. In the spatial transformer block, a Swin transformer block[27] is adopted to capture fine-grained spatial features with high efficiency. Local features flow fully in multi-head self-attention layers and densely interact with long-range global features. Finally, the output of the STB is remapped by the temporal decoder module, and its temporal scale can be rescaled to $T$. This decompression operation is implemented by two cascaded temporal decoders using convolutional layers with $3 \times 3$ kernels.

We used a linear combination of L1-norm loss and L2-norm loss as the loss function to optimize the parameters of SRDTrans, which shows better performance than L1-norm loss and improved convergence compared with L2-norm loss (Supplementary Fig. 23). We define the input substack filled with central pixels as $S_c$, the target substack filled with its vertically adjacent pixels as $S_v$ and the target substack filled with its horizontally adjacent pixels as $S_h$. The total loss consists of two pairs of training losses, which is defined as:

$$L_{ver} = \|F_{SRDTrans}(S_i) - S_v\|_2^2 + |F_{SRDTrans}(S_i) - S_v|_1,$$

$$L_{hor} = \|F_{SRDTrans}(S_i) - S_h\|_2^2 + |F_{SRDTrans}(S_i) - S_h|_1,$$

$$L_{total} = L_{ver} + L_{hor}.$$

where $L_{ver}$ and $L_{hor}$ denote the loss of the vertically and horizontally adjacent substacks, respectively.

### Training and inference

To achieve optimal performance, specific models were trained for stacks with different SNRs. One or more training stacks ($xy-t$ or $xy-z$) were divided into a specified number of 3D ($xy-t$) training pairs (6,000 by default). The batch size for all experiments was set to the number of graphics processing units (NVIDIA GeForce RTX 3090 for most cases) being used and the patch size was set to be $128 \times 128 \times 128$ pixels. All extracted training pairs were geometrically transformed by random flipping or rotation for eightfold data augmentation. The synergy of our lightweight architecture and data augmentation eliminates the risk of overfitting (Supplementary Fig. 24). The compression factor of each temporal encoder was set to 4. In the STB, we set the internal patch size to 7, the number of heads in the multi-headed self-attention block to 8 and the embedded feature channels to 128. For model optimization, we used the Adam optimizer and the exponential decay rate for the first moment was 0.9, the exponential decay rate for the second moment was 0.999 and the learning rate was 0.00001. PyTorch was used to construct the network and implement all operations. In the inference stage, the raw noisy data were not spatially subsampled and the model of the last training epoch was selected for final processing. The denoised result of each image stack was saved as a separate TIF file.

### Data simulation

Quantitative evaluations were performed on simulated data because noise-free images (GT) are available. To synthesize noise-free two-photon calcium imaging data, we used NAOMi[44], which can generate realistic calcium imaging data with high-fidelity tissue characteristics and fluorescence kinetics. Then we applied different levels of mixed Poisson–Gaussian noise to generate calcium imaging data of different imaging SNRs[6,18]. We also simulated data containing only Poisson or Gaussian noise to show the comparable denoising performance of SRDTrans on these two types of noise (Supplementary Fig. 25). To generate calcium imaging data of different sampling rate, we first synthesized images sampled at 30 Hz and 1 Hz, and the data of other sampling rates were obtained by extracting frames at different intervals. The image size for all simulated calcium imaging data was $490 \times 490$ pixels and the pixel size was 1.02 μm.

To generate realistic SMLM data, we first acquired reconstructed super-resolution images from the ShareLoc.XYZ platform (https://shareloc.xyz/)[42]. These images were experimentally obtained on a Nikon N-STORM microscope and contained densely distributed microtubules immuno-labeled with Alexa 647[54]. The tracks of all microtubules in a selected region of interest were extracted semi-automatically using

the JFilament plugin of ImageJ[55]. We then generated synthetic single-molecule-emission image stacks (GT images without noise) using the TestSTORM[36] simulator by loading the microtubule patterns from JFilament. All fluorescent molecules were linked on the microtubule pattern with a radius of 12.5 nm. For imaging parameters, the numerical aperture was 1.4 and the frame rate was 200 Hz (5 ms exposure time). The image size was 328 × 328 pixels and the pixel size was 30 nm. Noisy stacks were generated by applying mixed Poisson–Gaussian noise post hoc with a customized MATLAB script[6,18].

We synthesized moving objects with different moving speeds using the Modified National Institute of Standards and Technology (MNIST) dataset[56]. Each frame was defined as an image of 512 × 512 pixels with a black background and many bright moving digits. Each digit was an image patch (28 × 28 pixels) randomly extracted from the MNIST dataset, moved in a random direction, and appeared or disappeared only once. The total number of digits in the field of view (FOV) was 500. We first generated the data with a moving speed of 0.5 pixels per frame. The data with higher moving speeds were then generated by extracting frames at different intervals. The total frame number for all moving speeds was 5,000. The final experiment was implemented on 20 datasets with moving speeds from 0.5 to 10 pixels per frame. The sampling interval of the moving speed was 0.5 pixels per frame.

### SMLM imaging
The imaging samples (including the buffer solution and the sample holder) for the SMLM experiments were purchased from Standard Imaging Company (https://www.standardimaging.cn/standardsample?lang=en). The SMLM experiments were performed on a commercial microscope (Nikon N-STORM) equipped with laser sources of 405 nm and 640 nm, which were used for activation and excitation, respectively. A scientific complementary metal-oxide semiconductor camera (Hamamatsu Flash 4.0) was placed at the image plane to capture the emission signals. To mimic living-cell imaging, we used low excitation power to reduce phototoxicity and short exposure time to obtain images with low emitter density. The detailed settings are summarized in Supplementary Table 4.

### SMLM sample preparation
The Biologics Standards-Cercopithecus-1 (BSC-1) cell line purchased from Pricella Life Technology was used for SMLM imaging. BSC-1 cells were cultured in DMEM (Invitrogen, 11965-118) supplemented with 10% fetal bovine serum (Gibco, 16010-159). To prevent bacterial contamination, 100 μg ml$^{-1}$ penicillin and streptomycin (Invitrogen, 15140122) were added into the DMEM medium. Cells were grown under standard cell culture conditions (5% $CO_2$, humidified atmosphere at 37 °C). BSC-1 cells were plated on 1.5 glass-bottom dishes over 48 h before sample preparation. For cell passage, cells were washed with pre-warmed PBS (Life Technologies, 14190500BT) 3 times and digested with 25% trypsin (Gibco, 25200-056) for 30 s. BSC-1 l cell lines were tested for potential mycoplasma contamination (MycoAlert, Lonza) and all tests showed negative results. For immunofluorescence staining, cells were grown on 35 mm, 1.5 glass coverslips. We pre-treated glass-bottom dishes with fibronectin (Invitrogen, 33016015) for 1 h at 37 °C to increase cell adhesion. On the day of sample preparation, the cell density should be about 50–70%. Cells were fixed with 37 °C pre-warmed fixation buffer for 10 min, containing 4% paraformaldehyde (EMS) and 0.1% glutaraldehyde in PBS. Then the sample was washed three times with PBS. For quenching the background fluorescence, we incubated the cells with 2 ml 0.1% $NaBH_4$ solution in PBS for 7 min, optionally shaking on the shaker (<1 Hz). The sample was washed 3 times with 2 ml PBS and then incubated for 30 min in PBS containing 5% BSA (Jackson, 001-000-162) and 0.5% Triton X-100 (Fisher Scientific) at 37 °C. All antibodies were diluted in the 5% BSA + 0.5% triton solution described above. Next, we incubated the sample for 40 min with the appropriate dilution of primary antibodies: mouse anti-beta-tubulin

(E7, DSHB) at 25 °C. After primary antibodies incubation, the cells were washed 3 times with 2 ml PBS for 5 min. Secondary antibodies (AffiniPure Donkey Anti-mouse IgG, 715-005-150, Jackson Immuno Research) were incubated for 60 min with the appropriate dilutions of secondary antibodies (conjugated with Cy5) at 25 °C. After being washed 3 times with PBS, cells were fixed with post-fixation buffer for 10 min. The sample was stored at 4 °C in PBS and protected from light. Before imaging, we used an imaging buffer (STIBa-031, Standard Imaging Company) to replace PBS.

### SMLM reconstruction
The super-resolution SMLM images were reconstructed by the ThunderSTORM[37] Fiji plugin. For our experimentally obtained data, hard thresholding was performed to zero out those pixels with values smaller than a manually adjusted threshold to suppress the patterned noise of the camera. The detailed configuration is set as: the image filter was the wavelet filter (B-spline) with an order of 3 and a scale of 2; the algorithm for determining the approximate position of molecules was the local maximum algorithm; the subpixel localization is performed by the integrated Gaussian point-spread-function model with a fitting radius of 3 pixels; a fitting method of 'weighted least squares', and an initial sigma of 1.6 pixels. Both visualization images are generated by averaged shifted histogram with a magnification of 5. For better visualization, the single-molecule-emission images and reconstructed super-resolution images were rendered with pseudo-color and their contrast and brightness were manually adjusted.

### Mouse preparation and calcium imaging
All experiments involving mice were performed in accordance with the institutional guidelines for animal welfare and have been approved by the Animal Care and Use Committee of Tsinghua University. All mice were aged 8–12 weeks and were housed in cages (24 °C, 50% humidity) in groups of 1–5 under a reverse light cycle. Transgenic mice hybridized between Rasgrf2-2A-dCre mice and Ai148 (TIT2L-GC6f-ICL-tTA2)-D mice expressing Cre-dependent GCaMP6f genetically encoded calcium indicator were used for calcium imaging of neural circuits. Both male and female mice were used without randomization or blinding. Craniotomy surgeries were conducted to remove the skull and a coverslip was implanted on the craniotomy region for chronic imaging. Two-photon volumetric imaging of the mouse cortex was performed on head-fixed mice without anesthesia using a standard two-photon microscope controlled with ScanImage 5.7. The neural volume being recorded was located at the primary visual cortex with a depth of about 150–350 μm below the dura, and was scanned for 100 planes with an axial step of 2 μm. The whole imaging session lasted 30 min with a volume rate of 0.3 Hz.

### Three-dimensional visualization of neural activity
For volumetric calcium imaging, we used Imaris 9.0 (Oxford Instruments) to visualize the calcium activity of the neuronal population before and after denoising. Specifically, we imported the four-dimensional ($xyz-t$) data into Imaris, applied pseudo-color to the images, and then performed 3D rendering using the maximum intensity projection mode. The contrast and brightness were adjusted to make structures in the volume as clear as possible. All values for gamma correction were set to one.

### Method comparison
We compared the performance of SRDTrans with six baseline self-supervised methods: Noise2Noise[16], Noise2Void[19], Noise2Self[20], Probabilistic Noise2Void[21], Neighbor2Neighbor[22], DeepInterpolation[17] and DeepCAD[6,18]. These methods were all implemented by open-source codes released by the relevant papers. The denoising model of each method was trained and tested on the same datasets. For the methods designed for two-dimensional images, we split the time-lapse ($xy-t$)

image stack into a series of two-dimensional frames to match the input dimension. Training and inference were performed frame by frame. We followed the default training settings about network architectures and hyperparameters for all methods. Specifically, the model of DeepInterpolation was fine-tuned based on a public pre-trained model (pre-trained with 225,000 two-photon images of the Ai93 reporter line). Other methods were trained from scratch. The detailed settings of each method are listed in Supplementary Table 5.

### Evaluation metrics
We used several metrics to evaluate the performance of different denoising methods. For an image (or an image stack) $S_x$ and its GT $S_y$, the metrics are defined as follows.

SNR measures the pixel-level deviation between two images using the logarithmic decibel scale, which is formulated as

$$\text{SNR} = 10 \log_{10} \frac{\left\| S_y \right\|_2^2}{\left\| S_x - S_y \right\|_2^2}.$$

SSIM measures the similarity between two images on a perceptual level, including luminance, contrast and structure. The definition is

$$\text{SSIM} = \frac{(2\mu_x\mu_y + c_1)(2\sigma_{xy} + c_2)}{(\mu_x^2 + \mu_y^2 + c_1)(\sigma_x^2 + \sigma_y^2 + c_2)},$$

where $\{\mu_x, \mu_y\}$ and $\{\sigma_x, \sigma_y\}$ are the means and variances of $S_x$ and $S_y$, respectively. $\sigma_{xy}$ is the covariance of $S_x$ and $S_y$. The two constants $c_1$ and $c_2$ are defined as $c_1 = (k_1 L)^2$ and $c_2 = (k_2 L)^2$ with $k_1 = 0.01$, $k_2 = 0.03$ and $L = 65,535$.

The Jaccard index measures the proportion of correctly detected molecules in SMLM. The correctly localized fluorescent molecules are true positives (TP). The incorrectly localized molecules are false positives (FP) and the undetected molecules are false negatives (FN). The Jaccard index is formulated as:

$$\text{Jaccard} = 100 \frac{\text{TP}}{\text{TP} + \text{FP} + \text{FN}} \%.$$

The r.m.s.e. quantifies the mean difference between the localized positions ($P_x$) and GT positions ($P_y$) of all detected fluorescent molecules:

$$\text{r.m.s.e.} = \sqrt{\frac{1}{\text{TP}} \sum_{\text{TP}} \left\| P_y - P_x \right\|_2^2},$$

Efficiency ($E$) is a comprehensive metric combining the Jaccard index and r.m.s.e. to measure the performance of single-molecule localization[39]. It can simultaneously reflect the ability to detect molecules from images (measured by Jaccard) and the ability to precisely locate molecules (measured by r.m.s.e.), which is defined as:

$$\text{Efficiency} = 100 - \sqrt{(100 - \text{Jaccard})^2 + \alpha^2 \text{r.m.s.e.}^2},$$

where $\alpha = 1\,\text{nm}^{-1}$ controls the trade-off between Jaccard and r.m.s.e.

The Pearson correlation coefficient measures the similarity between a variable (images and fluorescence traces) and its GT, which is formulated as

$$R = \frac{E[(S_x - \mu_x)(S_y - \mu_y)]}{\sigma_x \sigma_y},$$

where $E$ represents the arithmetic mean. $\{\mu_x, \mu_y\}$ and $\{\sigma_x, \sigma_y\}$ are the means and variances of $S_x$ and $S_y$, respectively.

Logarithmic frequency distance (LFD) quantifies the spectral difference between two images in the frequency domain. For images with a size of $H \times W$ pixels, LFD is formulated as:

$$\text{LFD} = \log_{10} \left[ \frac{1}{HW} \left( \sum_{u=0}^{H-1} \sum_{v=0}^{W-1} \left\| F_{S_x}(u,v) - F_{S_y}(u,v) \right\|_2^2 \right) + 1 \right].$$

$F_{S_x}$ and $F_{S_y}$ are the discrete Fourier transform of $S_x$ and $S_y$, respectively. $u$ and $v$ are the pixel index in the frequency domain.

### Statistics and reproducibility
To ensure the reproducibility of the findings, we report the sample size and statistics in the legend and text of each experiment. All box plots are drawn in the standard Tukey box-and-whisker format. The upper and lower quartiles are represented by box bounds, and the lines in the boxes indicate the median. The lower whisker represents the minimum observed value, equal to the lower quartile minus 1.5× the interquartile range. The upper whisker the maximum observed value, equal to the upper quartile plus 1.5× the interquartile range. Results obtained through experimental or observational studies or statistical analysis of datasets can be reproduced with high reliability when the study is repeated. Representative images are shown in figures and similar results are achieved on all test samples. Experiments in Figs. 1d,e and 4a were repeated with 6,000 frames. Experiments in Figs. 2a and 3a,e were repeated with 24,000, 180,000 and 60,000 frames, respectively. Experiments in Figs. 4e and 5b were repeated with 1,000 and 548 frames, respectively.

### Reporting summary
Further information on research design is available in the Nature Portfolio Reporting Summary linked to this article.

## Data availability
Both simulated and experimentally obtained data of two-photon calcium imaging and single-molecule localization microscopy used in this work can be found at https://github.com/cabooster/SRDTrans/tree/main/datasets (refs. 57–60). Source data are provided with this paper.

## Code availability
The open-source Python code of SRDTrans is available at the Zenodo repository[61] and on GitHub (https://github.com/cabooster/SRDTrans).

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

## Acknowledgements

This research was supported by the National Natural Science Foundation of China (62088102, 62222508, 62071272) and National Key Research and Development Program of China (Project No. 2022YFB36066), in part by the Shenzhen Science and Technology Project under Grant (CJGJZD20200617102601004, JCYJ20220818101001004). This work was also supported by the Chinese Postdoctoral Foundation (BX2021159) and Shuimu Tsinghua Scholar Program. We thank H. Hao from Standard Imaging (Beijing) Biotechnology Co., Ltd for providing information about SMLM sample preparation.

## Author contributions

Q.D., H.W. and J.W. supervised this research. Q.D., H.W., J.W. and X.L. conceived and initiated this project. X.L., X.H. and X.C. designed detailed implementations and performed imaging experiments. X.H. and X.L. developed the Python code, performed simulations and processed relevant imaging data. J.F. prepared samples and provided models animals. Z.Z. gave critical support on the two-photon imaging system and imaging procedures. X.H., X.L. and X.C. analyzed the data, prepared figures and videos. X.L., X.H., X.C., J.F., Z.Z. and J.W. participated in discussions about the results and gave valuable advice. All authors participated in the drafting of the paper.

## Competing interests

The authors declare no competing interests.

## Additional information

**Correspondence and requests for materials** should be addressed to Jiamin Wu, Haoqian Wang or Qionghai Dai.

# Reporting Summary

## Statistics

For all statistical analyses, confirm that the following items are present in the figure legend, table legend, main text, or Methods section.

| n/a | Confirmed | |
|---|---|---|
| ☐ | ☒ | The exact sample size (*n*) for each experimental group/condition, given as a discrete number and unit of measurement |
| ☐ | ☒ | A statement on whether measurements were taken from distinct samples or whether the same sample was measured repeatedly |
| ☒ | ☐ | The statistical test(s) used AND whether they are one- or two-sided *Only common tests should be described solely by name; describe more complex techniques in the Methods section.* |
| ☒ | ☐ | A description of all covariates tested |
| ☒ | ☐ | A description of any assumptions or corrections, such as tests of normality and adjustment for multiple comparisons |
| ☐ | ☒ | A full description of the statistical parameters including central tendency (e.g. means) or other basic estimates (e.g. regression coefficient) AND variation (e.g. standard deviation) or associated estimates of uncertainty (e.g. confidence intervals) |
| ☒ | ☐ | For null hypothesis testing, the test statistic (e.g. *F*, *t*, *r*) with confidence intervals, effect sizes, degrees of freedom and *P* value noted *Give P values as exact values whenever suitable.* |
| ☒ | ☐ | For Bayesian analysis, information on the choice of priors and Markov chain Monte Carlo settings |
| ☒ | ☐ | For hierarchical and complex designs, identification of the appropriate level for tests and full reporting of outcomes |
| ☐ | ☒ | Estimates of effect sizes (e.g. Cohen's *d*, Pearson's *r*), indicating how they were calculated |

*Our web collection on statistics for biologists contains articles on many of the points above.*

## Software and code

Policy information about availability of computer code

| | |
|---|---|
| Data collection | The calcium imaging data were collected by a two-photon microscopes controlled by ScanImage 5.7 (Free release,Vidrio). The single-molecule localization microscopic imaging data were collected by a commercial Nikon N-STORM system equipped with laser sources of 405 nm and 640 nm. |
| Data analysis | Data simulation and result analysis were performed using custom Matlab (R2019b MathWorks) scripts. All deep learning models reported in this work were implemented with standard libraries of Python 3.6.0) with PyTorch (1.7.0, Facebook). All results of DeepCAD were obtained with the released code (https://github.com/cabooster/DeepCAD-RT). Single-molecule localization was implemented with the open-source ThunderSTORM plugin (v1.3) of Fiji (https://github.com/zitmen/thunderstorm/).The complete code of SRDTrans has been made publicly available at https://github.com/cabooster/SRDTrans. |

For manuscripts utilizing custom algorithms or software that are central to the research but not yet described in published literature, software must be made available to editors and reviewers. We strongly encourage code deposition in a community repository (e.g. GitHub). See the Nature Portfolio guidelines for submitting code & software for further information.

## Data

Policy information about availability of data

All manuscripts must include a data availability statement. This statement should provide the following information, where applicable:

- Accession codes, unique identifiers, or web links for publicly available datasets
- A description of any restrictions on data availability
- For clinical datasets or third party data, please ensure that the statement adheres to our policy

> We have no restrictions on data availability. Both the simulated and experimental data of single-molecule localization microscopy and two-photon calcium imaging in this work are available at https://github.com/cabooster/SRDTrans/tree/main/datasets. The MNIST dataset used for simulation is publicly available at http://yann.lecun.com/exdb/mnist/

## Human research participants

Policy information about studies involving human research participants and Sex and Gender in Research.

| | |
|---|---|
| Reporting on sex and gender | Not relevant. |
| Population characteristics | Not relevant. |
| Recruitment | Not relevant. |
| Ethics oversight | Not relevant. |

Note that full information on the approval of the study protocol must also be provided in the manuscript.

# Field-specific reporting

Please select the one below that is the best fit for your research. If you are not sure, read the appropriate sections before making your selection.

☒ Life sciences    ☐ Behavioural & social sciences    ☐ Ecological, evolutionary & environmental sciences

For a reference copy of the document with all sections, see nature.com/documents/nr-reporting-summary-flat.pdf

# Life sciences study design

All studies must disclose on these points even when the disclosure is negative.

| | |
|---|---|
| Sample size | The sample size(n) of each experiment is provided in the figure/table legends in the main manuscript and supplementary information files. |
| Data exclusions | No data was excluded from the analysis. |
| Replication | The number of repetitions for each experiment is provided in the figure/table legends in the main manuscript and supplementary information files. |
| Randomization | Not relevant, as there were no such experimental groups in this study. |
| Blinding | Not relevant, as there were no such experimental groups in this study. |

# Reporting for specific materials, systems and methods

We require information from authors about some types of materials, experimental systems and methods used in many studies. Here, indicate whether each material, system or method listed is relevant to your study. If you are not sure if a list item applies to your research, read the appropriate section before selecting a response.

## Materials & experimental systems

| n/a | Involved in the study |
|---|---|
| ☐ | ☒ Antibodies |
| ☐ | ☒ Eukaryotic cell lines |
| ☒ | ☐ Palaeontology and archaeology |
| ☐ | ☒ Animals and other organisms |
| ☒ | ☐ Clinical data |
| ☒ | ☐ Dual use research of concern |

## Methods

| n/a | Involved in the study |
|---|---|
| ☒ | ☐ ChIP-seq |
| ☒ | ☐ Flow cytometry |
| ☒ | ☐ MRI-based neuroimaging |

## Antibodies

| | |
|---|---|
| Antibodies used | Primary antibody: mouse anti-Beta-tubulin E7 (AB_2315513, E7, DSHB).<br>Secondary antibody: AffiniPure Donkey Anti-Mouse IgG (AB_2347508, 715-005-150, Jackson Immuno Research) conjugated with Cy5 (PA25001, GE Healthcare). |
| Validation | The antibodies have been extensively used in previous studies and the the statements related to antibody validation can be found at manufacturer' website(mouse anti-Beta-tubulin E7: https://dshb.biology.uiowa.edu/E7_2; AffiniPure Donkey Anti-Mouse IgG: https://www.jacksonimmuno.com/catalog/products/715-005-150). Labeled samples were purchased from Standard Imaging Company (https://www.standardimaging.cn/standardsample?lang−en). |

## Eukaryotic cell lines

Policy information about cell lines and Sex and Gender in Research

| | |
|---|---|
| Cell line source(s) | The Biologics Standards-Cercopithecus-1 (BSC-1) cell line purchased from Pricella Life Technology Co. LTD was used for STORM imaging. |
| Authentication | None of the cell lines were authenticated. |
| Mycoplasma contamination | Cell lines have been tested and were negative for mycoplasma contamination. |
| Commonly misidentified lines<br>(See ICLAC register) | No commonly misidentified lines were used. |

## Animals and other research organisms

Policy information about studies involving animals; ARRIVE guidelines recommended for reporting animal research, and Sex and Gender in Research

| | |
|---|---|
| Laboratory animals | For calcium imaging of neural circuits in the mouse brain, we used transgenic mice (male or female) hybridized between Rasgrf2-2A-dCre mice and Ai148 (TIT2L-GC6f-ICL-tTA2)-D mice expressing Cre-dependent GCaMP6f genetically encoded calcium indicator (GECI). All mice were aged 8-12 weeks and were housed in cages (24°C, 50% humidity) in groups of 1-5 under a reverse light cycle. |
| Wild animals | None |
| Reporting on sex | None |
| Field-collected samples | None |
| Ethics oversight | All experiments involving animals were performedin accordance with the institutional guidelines for animal welfare and have been approved by the Animal Care and Use Committee of Tsinghua University. |

Note that full information on the approval of the study protocol must also be provided in the manuscript.

