## [Peer Review File · Nature Computational Science]

Peer Review Information

Journal: Nature Computational Science

Manuscript Title: Spatial redundancy transformer for self-supervised fluorescence image denoising

Corresponding author name(s): Professor Qionghai Dai

Editorial Notes:

Reviewer Comments & Decisions:

Decision Letter, initial version:
--

Date: 26th August 23 11:11:13

Last Sent: 26th August 23 11:11:13

Triggered By: Ananya Rastogi

From: ananya.rastogi@nature.com

To: daiqh@tsinghua.edu.cn

BCC: ananya.rastogi@nature.com

Subject: Decision on Nature Computational Science manuscript NATCOMPUTSCI-23-0725

Message: ** Please ensure you delete the link to your author homepage in this e-mail if you wish to forward it to your co-authors. **

Dear Professor Dai,

Your manuscript "Spatial redundancy transformer for self-supervised fluorescence image denoising" has now been seen by 3 referees, whose comments are appended below. You will see that while they find your work of interest, they have raised points that need to be addressed before we can make a decision on publication.

The referees' reports seem to be quite clear. Naturally, we will need you to address all of the points raised.

While we ask you to address all of the points raised, the following points need to be substantially worked on:

- Please provide clear comparisons between DeepCAD and SRDTrans.
- As indicated by Reviewer #1, SRDTrans is primarily designed to address challenges in high-speed imaging. Please discuss if by foregoing temporal averaging, is there a

possibility that its SNR might be inferior to DeepCAD?

- Please show the impact on network performance if two or four pixels are randomly selected to form sub-stacks for training instead.
- Please discuss how inference performed in the actual experiment?
- Please ensure and verify the generalization ability of the network.
- As indicated by Reviewer #2, please provide further examples of the application of SRDTrans alternative modalities.
- As requested by Reviewer #3, please discuss whether orthogonal selection in the spatial domain is necessary.
- Please provide the description of the structure of the spatiotemporal transformer block.

Please use the following link to submit your revised manuscript and a point-by-point response to the referees' comments (which should be in a separate document to any cover letter):

[REDACTED]

** This url links to your confidential homepage and associated information about manuscripts you may have submitted or be reviewing for us. If you wish to forward this e-mail to co-authors, please delete this link to your homepage first. **

To aid in the review process, we would appreciate it if you could also provide a copy of your manuscript files that indicates your revisions by making use of Track Changes or similar mark-up tools. Please also ensure that all correspondence is marked with your Nature Computational Science reference number in the subject line.

In addition, please make sure to upload a Word Document or LaTeX version of your text, to assist us in the editorial stage.

To improve transparency in authorship, we request that all authors identified as 'corresponding author' on published papers create and link their Open Researcher and Contributor Identifier (ORCID) with their account on the Manuscript Tracking System (MTS), prior to acceptance. ORCID helps the scientific community achieve unambiguous attribution of all scholarly contributions. You can create and link your ORCID from the home page of the MTS by clicking on 'Modify my Springer Nature account'. For more information please visit <http://www.springernature.com/orcid>.

We hope to receive your revised paper within three weeks. If you cannot send it within this time, please let us know.

Best regards,

Ananya Rastogi, PhD
Senior Editor
Nature Computational Science

Reviewers comments:

Reviewer #1 (Remarks to the Author):

Fluorescence imaging is constrained by its limited photon budget, a critical factor that hampers its potential. Employing deep learning techniques for denoising has emerged as one of the promising solutions to this challenge. Several algorithms have been proposed in the past, including the author's own DeepCAD and DeepCAD-RT. These algorithms, as presented, rely heavily on the continuity of pixels between frames. Consequently, if the information within the image changes rapidly between frames, these algorithms tend to err. My tests have confirmed that they tend to force pixel values to approximate between frames. Beyond the author's contributions, several other algorithms, especially those based on CNNs, have been proposed in the field. Each of these, to varying degrees, has its limitations, such as the intricate parameter tuning required for non-self-supervised algorithms.

The primary motivation behind this paper appears to address the challenges faced by DeepCAD, especially in high-speed imaging scenarios. When there's significant variation between frames, an over-reliance on temporal pixel continuity becomes problematic. However, the spatial continuity of pixels remains a consistent feature. This paper capitalizes on this aspect, integrating a spatial redundancy sampling strategy with a lightweight spatio-temporal transformer architecture. SRDTrans holds significant promise for fluorescence microscopic image denoising, particularly in applications like single-molecule localization microscopy and two-photon volumetric calcium imaging.

However, the authors should provide clear delineation regarding scenarios where DeepCAD might be more appropriate versus those where SRDTrans shines. A direct comparison, backed by metrics like SNR, would be beneficial. It's evident that SRDTrans is primarily designed to address challenges in high-speed imaging, but by foregoing temporal averaging, is there a possibility that its SNR might be inferior to DeepCAD?

Further, if both DeepCAD and SRDTrans have their respective strengths, is there potential for an integrated algorithm? For biologists, choosing between algorithms can be daunting. For instance, in calcium imaging, where calcium signal response time hovers around 1 second, it would be valuable to know which algorithm to opt for based on the sampling rate relative to the response time.

Reviewer #1 (Remarks on code availability):

1. In the spatial redundancy sampling scheme, the authors randomly select three adjacent pixels from 2x2 small patches to form three sub-stacks. In these small patches, any two pixels can be considered as adjacent. What would be the impact on network performance if two or four pixels are randomly selected to form sub-stacks for training instead?

2. Why are there two MSA (Multi-Head Self-Attention) layers in the STB (Spatial Temporal Block)? Could their effects overlap, potentially rendering the second MSA

layer less effective? This question is especially pertinent considering there are eight heads in each MSA layer.

3. How is inference performed in the actual experiment? What type of data is used to train the network? Is it necessary to retrain the network when the experimental conditions (such as imaging targets or speeds) change?

4. Given the lightweight design of the network structure and the absence of a pooling layer, is there a risk of overfitting during network training? How can the generalization ability of the network be ensured and verified?

5. The L2 loss function tends to produce smoother network fitting results. Would employing the L1 loss function instead have a significant effect on the network's ability to fit more high-frequency information?

Reviewer #2 (Remarks to the Author):

This article demonstrates "SRDTrans", a self-supervised deep learning method of denoising time-lapse microscopy data. The novelty of creating a network that utilizes a spatiotemporal transformer architecture allows significant improvement over U-Net based architectures which by their nature tend to over-smooth high-frequency information. A related benefit is the light-weight nature of the transformer allows high-resolution information to be retained at a relatively low computational cost. Similarly, by employing a transformer model the network does not rely on similarities between temporally adjacent frames, and hence can be utilized on relatively fast-moving imaging modalities such as calcium-flux imaging or single molecule localization.

The authors show impressive denoising results on a variety of simulated and "real" (for lack of a better word) microscopy data. Not only demonstrating qualitative and quantitative improvements in the image, but also noting the improvement in the reliability down-stream analysis of the denoised data through SMLM point detection accuracy and calcium flux traces.

The article also directly compares SRDTrans to a range of other self-supervised denoising methods, mostly in the supplementary information. I'd be interested to see probabilistic noise2void (<https://www.frontiersin.org/articles/10.3389/fcomp.2020.00005/full>) tested alongside these other methods. While it does use a CNN and I suspect it will have the resolution degradation associated with its CNN architecture, it does take some temporal information into account. I grant that it does also require a separate training dataset and so may not technically fall within a self-supervised category, the collection of such a dataset is usually trivial to do at the time of data acquisition.

While I find the application of SRDTrans to both SMLM and multiphoton calcium imaging compelling, the authors note the applicability of the network to other microscopy techniques as there is no underlying assumptions about sample dynamics or imaging speed. Without wanting to place undue burden on the authors, I'd welcome any further examples of the application of SRDTrans alternative modalities –

I would think light sheet and/or SIM might be of particular interest to the field.

That said, I do not believe the work suffers from the absence of further examples as those presented are sufficiently impressive. I wish to commend the authors for their commitment to open science with the open-sourcing of their code which is well documented and easy to follow, as well as the availability of their training datasets and pre-trained models. I was able to test both training and inference on SMLM datasets with minimal alteration and look forward to further testing on my own datasets in the near future.

Reviewer #2 (Remarks on code availability):

The code is well documented on the github repo. I was able to test both training and inference on my own experimental data with minimal alteration.

I would suggest altering some of the filepath definitions in test.py and train.py to use `os.path.join` instead of string concatenation as I ran into some issues potentially due to OS differences. I also had to manually define the model `pth_name` variable in test.py as for a reason I couldn't determine the code was including "49" in the model name (I suspect due to the default checkpoint index but I didn't have time to debug more fully)

Reviewer #3 (Remarks to the Author):

This manuscript by Xinyang and colleagues presents a self-supervised-learning-based framework SRDtrans that removes noise from fluorescence time-lapse images. They provide a novel sampling strategy based on spatial redundancy to generate the training datasets to avoid the high dependency on quick imaging. A lightweight deep learning architecture is proposed to restore high-frequency information without producing over-smoothed structures. SRDtrans enables low-speed imaging of fast biological activities over a wide range of imaging SNR.

Image denoising is an important problem in image processing and computer vision. The spectral bias problem of CNN is a main limitation in self-supervised denoising tasks. This manuscript utilizes transformer to capture global spatiotemporal information can effectively solve this problem. Both synthetic and experimental results provided is impressive to convey it is a meaningful work.

Although the manuscript provides many illustrations, some statements and network characteristics remain unclear. And the statistics analyse should be further improved both in the figures (error bar should show in main SNR figure) , figure legends and tables .

Below I list my major comments:

1. In Fig 1a, spatial redundancy sampling strategy utilize orthogonal masks in spatial domain to generate the training datasets, and each timepoint (time domain) utilize different masks. I wonder whether the orthogonal selection in spatial domain is necessary. In each 2x2 cell of the raw images, can two neighboring pixels be

randomly chosen and categorized into two sub-images (just like Neighbor2Neighbor manner)? Or use the extra pixel in 2x2 cell and generate four sub-sampled images, one is selected as the training input, and other sub-stacks are designated as the corresponding training targets.

2. As one of the main innovation points, the structure of spatiotemporal transformer block needs detailed description. I had a glance on the provided codes, in SpatioTemporalTrans, it excute timeTrans first and then spatialTrans. It is better to describe this two step. In spatialTrans, it contains a SwinTransformerBlock, it may need a citation. In line 108-110 of the manuscript, what makes the proposed transformer lightweight? Less layers or less convolution kernels? For position embedding layer, does it require the input size been fixed, or can it deal with arbitrarily size?

3. In Supplementary Figure 5, large input temporal scale can provide better results in 30Hz data. This result is impressive. How about in 1Hz condition (I see in line 364, you have 1Hz synthesized data)? Will large temporal scale be harmful? It can help users to determine how many timepoints feed into the network is proper.

4. In line 279, it is said SRDTrans does not rely on any assumptions noise model. In the experiments, the authors added Mixed Gaussian-Poisson noise into the data, but I still wonder which noise (Gaussian or Poisson) have more severe influence on the SRDTrans results.

5. In line 247-248, the authors applied SRDTrans to volumetric recording, it is better to provide detailed process. Do they use spatial redundancy sampling to xyz-t data and change the network structure?

6. If possible, the experimentally obtained data needs a high-SNR reference to make the SRDTrans results convincing (e.g. in Fig 3b).

7. It is necessary for the authors to comment any failure case of SRDTrans or artifacts after recovery. Or to show the stability and generalization ability of SRDTrans.

8. The background of SRDTrans in Fig. 2a looks very clean, but when I applied both the self-trained model and the provided pretrained model (Pretrained_Model_for_noise_200Hz_2400frames_pxsize30nm_-0.05dBSNR_2400x328x328.pth) on cropped noisy data (noise_200Hz_2400frames_pxsize30nm_-0.05dBSNR_2400x328x328.tif), the results are not good. I think it is necessary to make a clarification about how to pre-process or post-process these data.

Some minor comments:

1. In line 118, it is better to point out which "typical deep layers", after the "STB" or the last layer.

2. In line 136-137, from Supplementary Video 1, DeepCAD provide better visual quality than SRDCNN, it is difficult to convince that "spatial redundancy sampling is more reasonable". I think Supplementary Table 3 can be a better evidence.

3. Line 293-295, which ablation study? Is fig. 3f a wrong citation?

4. Fig. 4f, Supplementary Figure 15a, Supplementary Table 2, how many samples are used to make statistics. It is better to add error bars in the main figure, i.e. Output SNR figure.

5. In the caption of Supplementary Table 3, "average SNR are shown in Supplementary Figure 14", but maybe Supplementary Figure 15. It is better to show average \pm SD.

Reviewer #3 (Remarks on code availability):

The code and datasets used in the manuscript can be easily access. The pretrained Calcium imaging model is useful, the codes are executable. It is appreciate that the authors provides sufficiently clear information and documentation for the code. However, the pretrained SMLM model seems not provide a nice result.

Author Rebuttal to Initial comments

Response letter

Editor's Remarks and Responses

Your manuscript "Spatial redundancy transformer for self-supervised fluorescence image denoising" has now been seen by 3 referees, whose comments are appended below. You will see that while they find your work of interest, they have raised points that need to be addressed before we can make a decision on publication.

The referees' reports seem to be quite clear. Naturally, we will need you to address all of the points raised.

We appreciate your time and effort in organizing the review of our manuscript. We are grateful for the reviewers' insightful comments and suggestions, which have substantially improved our manuscript. We have appended extensive experiments and statistical analyses according to the reviewers' comments. We believe the current manuscript has addressed all the points raised by the reviewers.

While we ask you to address all the points raised, the following points need to be substantially worked on:

1. Please provide clear comparisons between DeepCAD and SRDTrans.
2. As indicated by Reviewer #1, SRDTrans is primarily designed to address challenges in high-speed imaging. Please discuss if by foregoing temporal averaging, is there a possibility that its SNR might be inferior to DeepCAD?
3. Please show the impact on network performance if two or four pixels are randomly selected to form sub-stacks for training instead.
4. Please discuss how inference performed in the actual experiment?
5. Please ensure and verify the generalization ability of the network.
6. As indicated by Reviewer #2, please provide further examples of the application of SRDTrans alternative modalities.
7. As requested by Reviewer #3, please discuss whether orthogonal selection in the spatial domain is necessary.
8. Please provide the description of the structure of the spatiotemporal transformer block.

We have provided detailed responses to these points. All of them have been fully addressed with additional results and analyses. We also revised the manuscript according to the reviewers' comments to make these points clear to our readers.

Reviewer #1's Remarks and Responses

Fluorescence imaging is constrained by its limited photon budget, a critical factor that hampers its potential. Employing deep learning techniques for denoising has emerged as one of the promising solutions to this challenge. Several algorithms have been proposed in the past, including the author's own DeepCAD and DeepCAD-RT. These algorithms, as presented, rely heavily on the continuity of pixels between frames. Consequently, if the information within the image changes rapidly between frames, these algorithms tend to err. My tests have confirmed that they tend to force pixel values to approximate between frames. Beyond the author's contributions, several other algorithms, especially those based on CNNs, have been proposed in the field. Each of these, to varying degrees, has its limitations, such as the intricate parameter tuning required for non-self-supervised algorithms.

The primary motivation behind this paper appears to address the challenges faced by DeepCAD, especially in high-speed imaging scenarios. When there's significant variation between frames, an over-reliance on temporal pixel continuity becomes problematic. However, the spatial continuity of pixels remains a consistent feature. This paper capitalizes on this aspect, integrating a spatial redundancy sampling strategy with a lightweight spatio-temporal transformer architecture. SRDTrans holds significant promise for fluorescence microscopic image denoising, particularly in applications like single-molecule localization microscopy and two-photon volumetric calcium imaging.

We are grateful for your time and effort on the review of our manuscript. We have revised our manuscript and appended additional experiments and analyses according to your comments. We hope our responses can fully address the points you raised.

However, the authors should provide clear delineation regarding scenarios where DeepCAD might be more appropriate versus those where SRDTrans shines. A direct comparison, backed by metrics like SNR, would be beneficial. It's evident that SRDTrans is primarily designed to address challenges in high-speed imaging, but by foregoing temporal averaging, is there a possibility that its SNR might be inferior to DeepCAD?

Thanks for your insightful comment. SRDTrans and DeepCAD are based on different assumptions. SRDTrans was designed for data with spatial redundancy while DeepCAD was designed for data with temporal redundancy. **We have made quantitative comparisons in Fig. 4f**, which shows that spatial redundancy (SRDTrans) is better at low imaging speed, and temporal redundancy (DeepCAD) is better at high imaging speed. The superiority of SRDTrans over DeepCAD at high imaging speeds is attributed to the transformer architecture. If the network architecture of DeepCAD is replaced with that of SRDTrans, its denoising performance will surpass SRDTrans. Thus, determining which method to use should depend on which kind of redundancy is more sufficient in the data. However, at present, we can only qualitatively delineate the scenarios where each of them is appropriate, rather than giving a demarcation point quantitatively. Combined with Fig. 4f, we have provided extensive discussions (in the 'Discussion' section) to make this point clear.

Throughout our work, temporal averaging has never been used. Those low-frame-rate

calcium imaging data were obtained by extracting frames at different intervals. When comparing the performance of SRDTrans and DeepCAD at different frame rates, all other factors except the frame rate remained the same. It's inappropriate to reduce the frame rate by temporal averaging, since the SNR of the data will be improved. We hope we've understood your concern and addressed it properly.

Fig. 4f

Further, if both DeepCAD and SRDTrans have their respective strengths, is there potential for an integrated algorithm? For biologists, choosing between algorithms can be daunting. For instance, in calcium imaging, where calcium signal response time hovers around 1 second, it would be valuable to know which algorithm to opt for based on the sampling rate relative to the response time.

DeepCAD and SRDTrans were designed to exploit different aspects of redundancy. DeepCAD is more suitable for data with temporal redundancy and SRDTrans is more suitable for data with spatial redundancy. Thus, developing an integrated algorithm requires evaluating which aspect of redundancy is more adequate. However, without ground truth, it is difficult to quantify the spatial and temporal redundancy. Those commonly used metrics such as PSNR/SSIM/Pearson Correlation cannot reflect the similarity between spatially or temporally adjacent images because of noise contamination. Although the relationship between the indicator kinetics and sampling rate seems to provide a rough reference, this method could be misleading since the indicator kinetics vary significantly in different model organisms even for the same kind of indicator, not to mention a wide range of other biological activities. That's why we leave the choice between DeepCAD and SRDTrans empirical. We think quantifying data redundancy by image similarity is more feasible than by biophysics and imaging parameters. Using more advanced and sophisticated methods, it is possible to develop an integrated algorithm that can automatically choose between DeepCAD and SRDTrans. This is a valuable future work that we would like to explore to realize a general denoising method.

Remarks on code availability

1. In the spatial redundancy sampling scheme, the authors randomly select three adjacent pixels from 2x2 small patches to form three sub-stacks. In these small patches, any two pixels can be considered as adjacent. What would be the impact on network performance if two or four pixels are randomly selected to form sub-stacks for training instead?

Thanks for raising this question. To answer your question about the sampling scheme, we have conducted additional experiments to compare the performance of different sampling schemes both on calcium imaging and SMLM data. Here, four sampling schemes are investigated, including (1) diagonal sampling that only randomly selects diagonal pixels, (2) random sampling that selects horizontally or vertically or diagonally adjacent pixels, (3) neighboring sampling that selects horizontally or vertically adjacent pixels, (4) orthogonal sampling that simultaneously selects horizontally and vertically adjacent pixels. Quantitative results are shown in **Supplementary Fig. 2** (a snapshot is shown below).

Among them, diagonal sampling has the worst performance because diagonal pixels have the longest distance in a 2x2 patch and their similarity is the lowest. Using diagonal pixels for training will bring inferior performance. That's why we exclude diagonal pixels when generating training pairs. **The proposed orthogonal sampling has the best performance** since it can isotropically learn horizontal and vertical correlations and avoid using diagonal pixels. The comparison between orthogonal sampling and neighboring sampling can verify the necessity and superiority of the proposed orthogonal sampling scheme.

Supplementary Fig. 2

2. Why are there two MSA (Multi-Head Self-Attention) layers in the STB (Spatial Temporal Block)? Could their effects overlap, potentially rendering the second MSA layer less effective? This question is especially pertinent considering there are eight heads in each MSA layer.

Thanks for your questions. In theory and practice, multiple cascaded MSA layers do

not overlap with each other. They have independent parameters and would learn different levels of features. Most papers about transformers use multi-layer MSA to increase the diversity of features, just like the multiple cascaded convolutional layers in CNNs.

In the STB, we stack two layers of MSA both in the temporal transformer block and the spatial transformer block for different reasons. In the temporal transformer block, we empirically use two MSA layers to extract global temporal features. Generally, two MSA layers can learn representations better and capture high-level features with richer semantic information than a single MSA layer. In the spatial transformer block, we follow the standard Swin Transformer [ICCV 2021: 10012-10022] architecture to reduce the computational burden of self-attention blocks. The first layer is window-based MSA (W-MSA), and the second layer is shifted window-based MSA (SW-MSA). The SW-MSA brings greater efficiency by limiting self-attention computation to non-overlapping local windows while also allowing for cross-window connection.

3. (1) How is inference performed in the actual experiment? (2) What type of data is used to train the network? (3) Is it necessary to retrain the network when the experimental conditions (such as imaging targets or speeds) change?

(1) In actual experiments, the raw noisy data is both the training set and the test set. Pre-trained models will be directly applied to the raw noisy data without spatial sampling. (2) One or more TIF stacks (xy-t or xy-z) containing the raw noisy images were used for training. (3) Training specific models for specific data is the most reliable way to use SRDTrans, as well as other deep-learning-based methods for fluorescence image restoration. Thus, a new model should be trained to ensure optimal results when the experimental conditions change. A comprehensive investigation of the generalization ability has been summarized in **Supplementary Fig. 8** and can be found in the response to your next question. We have added these details in the revised manuscript to make these points clear.

4. Given the lightweight design of the network structure and the absence of a pooling layer, is there a risk of overfitting during network training? How can the generalization ability of the network be ensured and verified?

We appreciate your insightful comment. Overfitting and generalization are issues that all deep learning methods must consider. To answer your question about overfitting and generalization ability, we have conducted quantitative experiments and the results and conclusions are summarized below.

To evaluate overfitting, we trained SRDTrans for 40 epochs using simulated calcium imaging data. The model of each epoch was then tested and compared with the ground truth qualitatively (representative images) and quantitatively (SNR). The results are summarized in **Supplementary Fig. 24** (a snapshot is shown below), which shows that SRDTrans can keep stable convergence and no overfitting occurs. Such convergence

stability is attributed to the synergy of our lightweight architecture and data augmentation. Generally, training for 20 epochs (the default setting of our code) is adequate to make the model converge and obtain good performance. Thus, the users don't need to worry about overfitting when using SRDTrans.

Supplementary Fig. 24

To evaluate the generalization ability of our method, we conducted a series of cross-dataset and cross-modality experiments, including (1) inference with models trained on different SNRs (e.g., processing low-SNR data with a model trained on high-SNR data) and (2) inference with models trained on different imaging modalities (e.g., processing calcium imaging data with a model trained on SMLM data). Qualitative (representative images) and quantitative (SNR) results are summarized in Supplementary Fig. 8 (a snapshot is shown below). As expected, the best performance is obtained when the model is trained on matched SNR and imaging modality. If a pre-trained model is applied to different imaging modalities or SNRs, the denoising performance will degrade. Thus, a new model should be trained to ensure optimal results when the imaging parameter, modality, and sample change.

Since generalizability is a problem that has not been solved so far in deep learning, we believe training specific models for specific data is the most reliable way to use deep learning for image denoising in fluorescence imaging.

Supplementary Fig. 8

5. The L2 loss function tends to produce smoother network fitting results. Would employing the L1 loss function instead have a significant effect on the network's ability to fit more high-frequency information?

Thanks for this question. To figure out which loss function is better, we trained a model for each loss function (L1, L2, L1+L2) using the same dataset and hyperparameters. Their convergence curves are shown in Supplementary Fig. 23, which indicates that the performance of L1 loss is inferior to that of L2 loss and L1+L2 loss. Although L1 loss and L1+L2 loss have comparable performance, we observe that the convergence of L1+L2 loss is more stable. We have carefully checked the denoised images of these loss functions and did not find that L1 loss can restore more high-frequency structures. Taking denoising performance and convergence stability into consideration, we think L1+L2 loss is the best choice.

Supplementary Fig. 23

Reviewer #2's Remarks and Responses

This article demonstrates “SRDTrans”, a self-supervised deep learning method of denoising time-lapse microscopy data. The novelty of creating a network that utilizes a spatiotemporal transformer architecture allows significant improvement over U-Net based architectures which by their nature tend to over-smooth high-frequency information. A related benefit is the light-weight nature of the transformer allows high-resolution information to be retained at a relatively low computational cost. Similarly, by employing a transformer model the network does not rely on similarities between temporally adjacent frames, and hence can be utilized on relatively fast-moving imaging modalities such as calcium-flux imaging or single molecule localization.

The authors show impressive denoising results on a variety of simulated and “real” (for lack of a better word) microscopy data. Not only demonstrating qualitative and quantitative improvements in the image, but also noting the improvement in the reliability down-stream analysis of the denoised data through SMLM point detection accuracy and calcium flux traces.

We really appreciate your time and effort on the review of our manuscript. We have revised our manuscript and appended additional experiments according to your comments, including the comparison with Probabilistic Noise2Void and applications on more imaging modalities. These results substantially improve our work and demonstrate the superiority and wide applicability of our method. We hope our responses can fully address the points you raised.

The article also directly compares SRDTrans to a range of other self-supervised denoising methods, mostly in the supplementary information. I'd be interested to see probabilistic noise2void (<https://www.frontiersin.org/articles/10.3389/fcomp.2020.00005/full>) tested alongside these other methods. While it does use a CNN and I suspect it will have the resolution degradation associated with its CNN architecture, it does take some temporal information into account. I grant that it does also require a separate training dataset and so may not technically fall within a self-supervised category, the collection of such a dataset is usually trivial to do at the time of data acquisition.

Thanks for this comment. We have compared SRDTrans with Probabilistic Noise2Void (PN2V) using simulated SMLM data. Qualitative and quantitative results of PN2V are summarized in **Supplementary Fig. 10** (a snapshot is shown below) and presented alongside other methods. The comprehensive comparison shows that the denoising performance of PN2V is better than Noise2Self and Noise2Void, while inferior to Neighbor2Neighbor, DeepCAD and SRDTrans. The proposed SRDTrans has a significant superiority over PN2V and can restore more structures.

Supplementary Fig. 10

While I find the application of SRDTrans to both SMLM and multiphoton calcium imaging compelling, the authors note the applicability of the network to other microscopy techniques as there is no underlying assumptions about sample dynamics or imaging speed. Without wanting to place undue burden on the authors, I'd welcome any further examples of the application of SRDTrans alternative modalities – I would think light sheet and/or SIM might be of particular interest to the field.

Thanks for this valuable suggestion. To demonstrate the wide applicability of our method, we extend SRDTrans to another two very commonly used imaging modalities, i.e., **confocal microscopy** and **light sheet microscopy**. The results are shown in **Supplementary Fig. 20** (a snapshot is shown below), which indicates that SRDTrans still has excellent performance on the two imaging modalities. Although we could only use our existing data and publicly available data because of the limited time for revision, we believe the two examples can fully verify the applicability of SRDTrans on a wide range of imaging techniques, and arouse the interest of more readers.

Supplementary Fig. 20

That said, I do not believe the work suffers from the absence of further examples as those presented are sufficiently impressive. I wish to commend the authors for their commitment to open science with the open-sourcing of their code which is well documented and easy to follow, as well as the availability of their training datasets and pre-trained models. I was able to test both training and inference on SMLM datasets with minimal alteration and look forward to further testing on my own datasets in the near future.

We really appreciate your recognition of our work and code. We will continue our efforts to provide good open-source tools to the community.

Remarks on code availability

The code is well documented on the GitHub repo. I was able to test both training and inference on my own experimental data with minimal alteration.

I would suggest altering some of the filepath definitions in test.py and train.py to use `os.path.join` instead of string concatenation as I ran into some issues potentially due to OS differences. I also had to manually define the model `pth_name` variable in test.py as for a reason I couldn't determine the code was including "49" in the model name (I suspect due to the default checkpoint index but I didn't have time to debug more fully)

Thanks for this suggestion. We have modified the command for filepath operation and optimized the code systematically. We will continue to update the code to address other issues that may arise.

Reviewer #3's Remarks and Responses

This manuscript by Xinyang and colleagues presents a self-supervised-learning-based framework SRDtrans that removes noise from fluorescence time-lapse images. They provide a novel sampling strategy based on spatial redundancy to generate the training datasets to avoid the high dependency on quick imaging. A lightweight deep learning architecture is proposed to restore high-frequency information without producing over-smoothed structures. SRDtrans enables low-speed imaging of fast biological activities over a wide range of imaging SNR.

Image denoising is an important problem in image processing and computer vision. The spectral bias problem of CNN is a main limitation in self-supervised denoising tasks. This manuscript utilizes transformer to capture global spatiotemporal information can effectively solve this problem. Both synthetic and experimental results provided is impressive to convey it is a meaningful work.

Although the manuscript provides many illustrations, some statements and network characteristics remain unclear. And the statistics analysis should be further improved both in the figures (error bar should show in main SNR figure), figure legends and tables.

We are grateful for your time and effort on the review of our manuscript. We have revised our manuscript and appended additional experiments and analyses to address your concerns. We have also refined our statements and statistics. Your insightful comments and valuable suggestions have substantially improved our manuscript. We hope our responses can fully address the points you raised.

Major comments

1. In Fig 1a, spatial redundancy sampling strategy utilize orthogonal masks in spatial domain to generate the training datasets, and each timepoint (time domain) utilize different masks. I wonder whether the orthogonal selection in spatial domain is necessary. In each 2x2 cell of the raw images, can two neighboring pixels be randomly chosen and categorized into two sub-images (just like Neighbor2Neighbor manner)? Or use the extra pixel in 2x2 cell and generate four sub-sampled images, one is selected as the training input, and other sub-stacks are designated as the corresponding training targets.

Thanks for raising this question. To answer your question about the sampling scheme, **we have appended additional experiments to compare the performance of different sampling schemes both on calcium imaging and SMLM data.** Here, four sampling schemes are investigated, including (1) diagonal sampling that only randomly selects diagonal pixels, (2) random sampling that selects horizontally or vertically or diagonally adjacent pixels, (3) neighboring sampling that selects horizontally or vertically adjacent pixels, (4) orthogonal sampling that simultaneously selects horizontally and vertically adjacent pixels. Quantitative results are shown in **Supplementary Figure 2** (a snapshot is shown below).

Among them, diagonal sampling has the worst performance because diagonal pixels have the longest distance in a 2x2 patch and their similarity is the lowest. Using

diagonal pixels for training will bring inferior performance. That's why we exclude diagonal pixels when generating training pairs. **The proposed orthogonal sampling has the best performance** since it can isotropically learn horizontal and vertical correlations and avoid using diagonal pixels. The comparison between orthogonal sampling and neighboring sampling can verify the necessity and superiority of the proposed orthogonal sampling method.

Supplementary Figure 2

2. As one of the main innovation points, the structure of spatiotemporal transformer block needs detailed description. (1) I had a glance on the provided codes, in SpatioTemporalTrans, it excute timeTrans first and then spatialTrans. It is better to describe these two steps. In spatialTrans, it contains a SwinTransformerBlock, it may need a citation. (2) In line 108-110 of the manuscript, what makes the proposed transformer lightweight? Less layers or less convolution kernels? (3) For position embedding layer, does it require the input size been fixed, or can it deal with arbitrarily size?

We appreciate your valuable comment about the network architecture that help us improve our manuscript. Your questions are answered point-by-point:

(1) **To better describe the network, we have appended Supplementary Fig. 22** to visualize the architecture of the spatiotemporal transformer block (STB). Specifically, the STB contains a temporal transformer block (timeTrans in the code) and a spatial transformer block (spatialTrans in the code). In the temporal transformer block, the image patches are flattened into one-dimensional vectors and input into the position embedding layer, where spatial concatenation and linear transformation are performed. Then, two self-attention blocks are cascaded to extract temporal correlations inside the data. In the spatial transformer block, a Swin Transformer blocks are adopted to capture fine-grained spatial features. **We have made detailed descriptions of the STB architecture in "Methods". The relevant papers have been cited (Ref. 37-40).**

Supplementary Fig. 22

- (2) We use the following three ways to make the proposed transformer lightweight. (i) As shown in Fig. 1c, we design two temporal encoders before the STB module to reduce the temporal scale of the input. Each temporal encoder can compress the temporal scale of the input sub-stack by 4-fold using a convolutional layer with 3×3 kernels. (ii) In the spatial transformer block, we apply the window shifting mechanism from the Swin Transformer to compute window attention within image patches rather than the entire image, effectively reducing the computational complexity from quadratic to linear. (iii) Our architecture uses less layers. There are only two layers in the temporal and spatial transformer block, while a typical Swin Transformer often stacks tens of it.
- (3) The position embedding layer does not have fixed parameters or specific settings for the input size. Therefore, it does not require the input size to be fixed and can deal with arbitrary size.

3. In Supplementary Figure 5, large input temporal scale can provide better results in 30Hz data. This result is impressive. How about in 1Hz condition (I see in line 364, you have 1Hz synthesized data)? Will large temporal scale be harmful? It can help users to determine how many timepoints feed into the network is proper.

We repeat the experiment in the previous Supplementary Figure 5 (now it is Supplementary Figure 6) with 1-Hz data and find that the conclusion remains the same. Qualitative and quantitative results in Supplementary Figure 7 show that better denoising performance can be obtained at larger temporal scales regardless of the imaging speed. This property is attributed to the global receptive fields of our 3D-Transformer architecture. The more information that is input, the more information can be perceived and utilized by the network. Thus, if the computing device allows and there is enough data, the temporal scale should be as large as possible. However,

this effect can saturate. As shown in Fig. 1g, the output SNR will not increase much after the temporal scale exceeds 256. After balancing the device/data burden and denoising performance, we empirically set the default value of the temporal scale to 128, which can satisfy the vast majority of needs.

Supplementary Figure 7

4. In line 279, it is said SRDTrans does not rely on any assumptions on noise model. In the experiments, the authors added Mixed Gaussian-Poisson noise into the data, but I still wonder which noise (Gaussian or Poisson) have more severe influence on the SRDTrans results.

Mixed Gaussian-Poisson (MPG) noise is the general form of noise in fluorescence microscopy [IEEE Trans. Image Process. 27(8): 3842-3856 (2018), CVPR 2019, 11710-11718, Nat. Methods 18(6): 678-687 (2021), Nat. Biotechnol. 41(2): 282-292 (2023)]. The Poisson component (shot noise) is mainly caused by the quantum nature of photon

detection, while the Gaussian component is mainly caused by charge-voltage conversion. Benefiting from advanced semiconductor fabrication and sensor cooling techniques, the Gaussian component can be effectively suppressed. Thus, we used MPG noise with a dominant Poisson component as the noise model, which is consistent with the basic principle of photodetection in fluorescence imaging.

To answer your question about which noise (Gaussian or Poisson) has more severe influence on SRDTrans, we generate simulated calcium imaging data containing different noises but with the same SNR, so that the influence of input SNR can be excluded. Quantitative results and example images are summarized in **Supplementary Figure 25**, which shows that **SRDTrans has comparable denoising performance on Gaussian and Poisson noise**. Specifically, the output SNR of Gaussian noise is slightly higher (~3.6%) than that of Poisson noise.

Supplementary Figure 25

5. In line 247-248, the authors applied SRDTrans to volumetric recording, it is better to provide detailed process. Do they use spatial redundancy sampling to xyz-t data and change the network structure?

Thanks for this comment. For the denoising of volumetric calcium imaging data, we extracted all the frames of each imaging plane and reorganized them into a separate time-lapse stack (i.e., converting xyz-t data to xyt-z data). The time-lapse stacks of all imaging planes were used for network training. Spatial redundancy sampling was performed on xy-t stacks and the network architecture was not changed. We have added these details in the main text to clarify the processing pipeline of volumetric

calcium imaging data.

6. If possible, the experimentally obtained data needs a high-SNR reference to make the SRDTrans results convincing (e.g. in Fig 3b).

Thanks for this suggestion. Since the SMLM experiment was conducted on a commercial microscope, we cannot modify it to capture a synchronized high-SNR reference. Alternatively, we have verified the performance of SRDTrans on experimentally obtained calcium imaging data with synchronized high-SNR (10-fold fluorescence photons) reference (see **Supplementary Figure 15**). The result shows that those structures and dynamics swamped by noise can be restored authentically. We hope this supplementary experiment can address your concern.

Supplementary Figure 15

7. It is necessary for the authors to comment any failure case of SRDTrans or artifacts after recovery. Or to show the stability and generalization ability of SRDTrans.

Thanks for the valuable comment. To answer your question, we performed a series of cross-dataset and cross-modality experiments, including (1) inference with models trained on different SNRs (e.g., processing low-SNR data with a model trained on high-SNR data) and (2) inference with models trained on different imaging modalities (e.g., processing calcium imaging data with a model trained on SMLM data). Qualitative (representative images) and quantitative (SNR) results are summarized in **Supplementary Fig. 8** (a snapshot is shown below). As expected, the

best performance is obtained when the model is trained on matched SNR and imaging modality. If a pre-trained model is applied to different imaging modalities or SNRs, the denoising performance will degrade. Thus, a new model should be trained to ensure optimal results when the imaging parameter, modality, and sample change. Some failure cases caused by data variance have been presented in Supplementary Fig. 8c,8d. Since generalizability is a problem that has not been solved so far in deep learning, we believe training specific models for specific data is the most reliable way to use deep learning for image denoising in fluorescence imaging.

Supplementary Figure 8

8. The background of SRDTrans in Fig. 2a looks very clean, but when I applied both the self-trained model and the provided pretrained model (Pretrained_Model_for_noise_200Hz_2400frames_pxsize30nm_-0.05dBSNR_24000x328x328.pth) on cropped noisy data (noise_200Hz_2400frames_pxsize30nm_-0.05dBSNR_24001x328x328.tif), the results are not good. I think it is necessary to make a clarification about how to pre-process or post-process these data.

Thanks for scrutinizing the code and model. We have checked the code and model carefully and found that they were not the optimal version. We have updated them to ensure that our users can obtain optimal results. For the results in Fig. 2a, we did not

have additional pre-processing or post-processing. The only treatment was pseudo-color rendering and adjusting the contrast and brightness for better visualization.

Minor comments

1. In line 118, it is better to point out which “typical deep layers”, after the “STB” or the last layer.

In SRDTrans, the visualized feature maps are from the last layer of STB. In SRDCNN, the visualized feature maps are from the last layer of the 3D U-Net. We have added these details to make the description clear.

2. In line 136-137, from Supplementary Video 1, DeepCAD provide better visual quality than SRDCNN, it is difficult to convince that “spatial redundancy sampling is more reasonable”. I think Supplementary Table 3 can be a better evidence.

Thanks for reminding. We have cited Supplementary Table 3 here.

3. Line 293-295, which ablation study? Is fig. 3f a wrong citation?

We apologize for this typo. Here it should be Fig.4f. We have corrected the figure number and check the whole manuscript to avoid similar mistakes.

4. Fig. 4f, Supplementary Figure 15a, Supplementary Table 2, how many samples are used to make statistics. It is better to add error bars in the main figure, i.e. Output SNR figure.

Thanks for your careful review. We have added the sample size for all statistical analyses, as well as added error bars for all output SNR figures throughout the manuscript (Fig. 1g, 2c, 4d, 4f; Supplementary Fig. 2, 3, 6, 7, 8, 15, 18, 19, 24).

5. In the caption of Supplementary Table 3, “average SNR are shown in Supplementary Figure 14”, but maybe Supplementary Figure 15. It is better to show average \pm SD.

We apologize for this typo. Here it should be Supplementary Fig. 17 (the previous Supplementary Fig. 13). We have corrected the figure number and changed all SNR values into the form of mean \pm s.d. in Supplementary Table 3.

Frame rate		0.1 Hz	0.3 Hz	1 Hz	3 Hz	10 Hz	30 Hz
SRDCNN	3D-U-Net	17.66	18.04	18.67	20.61	21.64	22.41
	Spatial redundancy	± 0.43	± 0.56	± 0.64	± 0.33	± 0.28	± 0.37
DeepCAD	3D-U-Net	12.77	13.10	18.44	21.34	22.72	23.61
	Temporal redundancy	± 2.01	± 1.61	± 0.87	± 0.25	± 0.22	± 0.22
SRDTrans	3D-Transformer	19.59	19.62	20.83	22.44	23.76	25.08
	Spatial redundancy	± 0.37	± 0.31	± 0.36	± 0.30	± 0.30	± 0.23

Supplementary Table 3

Remarks on code availability

The code and datasets used in the manuscript can be easily access. The pretrained Calcium imaging model is useful, the codes are executable. It is appreciate that the authors provides sufficiently clear information and documentation for the code. However, the pretrained SMLM model seems not provide a nice result.

We really appreciate your recognition of our code and datasets. We will continue our efforts to provide good open-source tools to the community.

Thanks for reminding us of the SMLM model. We have checked the model and found that it was not trained by our final code. We have updated the model and code to ensure that our users can obtain optimal results. We will continue to update the code to address other issues that may arise.

Decision Letter, first revision:

Date: 6th October 23 12:51:58
Last Sent: 6th October 23 12:51:58
Triggered By: Ananya Rastogi
From: ananya.rastogi@nature.com
To: daiqh@tsinghua.edu.cn
CC: computacionalscience@nature.com
BCC: ananya.rastogi@nature.com
Subject: AIP Decision on Manuscript NATCOMPUTSCI-23-0725A
Message: Our ref: NATCOMPUTSCI-23-0725A

6th October 2023

Dear Dr. Dai,

Thank you for submitting your revised manuscript "Spatial redundancy transformer for self-supervised fluorescence image denoising" (NATCOMPUTSCI-23-0725A). It has now been seen by the original referees and their comments are below. The reviewers find that the paper has improved in revision, and therefore we'll be happy in principle to publish it in Nature Computational Science, pending minor revisions to satisfy the referees' final requests and to comply with our editorial and formatting guidelines.

TRANSPARENT PEER REVIEW

Nature Computational Science offers a transparent peer review option for original research manuscripts. We encourage increased transparency in peer review by publishing the reviewer comments, author rebuttal letters and editorial decision letters if the authors agree. Such peer review material is made available as a supplementary peer review file. **Please remember to choose, using the manuscript system, whether or not you want to participate in transparent peer review.**

Thank you again for your interest in Nature Computational Science. Please do not hesitate to contact me if you have any questions.

Sincerely,

Ananya Rastogi, PhD
Senior Editor
Nature Computational Science

ORCID

Reviewer #1 (Remarks to the Author):

The authors properly solved all the issues with additional results and analyses.

Reviewer #1 (Remarks on code availability):

The authors properly solved all the issues. The code is fine.

Reviewer #2 (Remarks to the Author):

The additional experiments and data greatly improve the manuscript and fully address my points raised at first review.

Reviewer #3 (Remarks to the Author):

I commend the authors for the extensive responses. I do see that authors have put efforts in showing some experimental support regarding the scientific concerns raised by me. The revised manuscript includes new experiments and clearer explanations that better delineate the advantages of the SRDTrans.

There remains one problem. I cannot check the SMLM model. When I test the model, the code showed error 'size mismatch for encoders.0.conv_net.SingleConv1.Conv3d.weight'. It seems the code have been modified, the channel numbers are increased. I hope the authors ensures the consistency of the code.

Reviewer #3 (Remarks on code availability):

I downloaded the model from Zenodo, the Calcium imaging model can be used, but the SMLM model showed "size mismatch" error. As the authors said they will continue to update the code to address other issues that may arise. I think make sure the SMLM model correct is neceaaasy.

Author Rebuttal, first revision:

response letter

Reviewer #3's Remarks and Responses

I commend the authors for the extensive responses. I do see that authors have put efforts in showing some experimental support regarding the scientific concerns raised by me. The revised manuscript includes new experiments and clearer explanations that better delineate the advantages of the SRDTrans.

There remains one problem. I cannot check the SMLM model. When I test the model, the code showed error 'size mismatch for encoders.0.conv_net.SingleConv1.Conv3d.weight'. It seems the code have been modified, the channel numbers are increased. I hope the authors ensures the consistency of the code.

We appreciate the reviewer for recognizing our revision.

We have checked the latest code and model and found that the SMLM model can work well. The issue of the reviewer may be caused by the reviewer not using the latest version of the code. Another possible reason is that the "--patch_t" used for testing is not consistent with that used for training. We have released our code on GitHub. The users can report issues on it at any time so that we can help them address issues that may arise.

Final Decision Letter:

Date: 7th November 23 14:43:25

Last Sent: 7th November 23 14:43:25

Triggered By: Ananya Rastogi

From: ananya.rastogi@nature.com

To: daiqh@tsinghua.edu.cn

BCC: fernando.chirigati@us.nature.com,rjsart@springernature.com,ananya.rastogi@nature.com,rjproduction@springernature.com,computationalscience@nature.com

Subject: Decision on Nature Computational Science manuscript NATCOMPUTSCI-23-0725B

Message: Dear Professor Dai,

We are pleased to inform you that your Article "Spatial redundancy transformer for self-supervised fluorescence image denoising" has now been accepted for publication in Nature Computational Science.

Once your manuscript is typeset, you will receive an email with a link to choose the appropriate publishing options for your paper and our Author Services team will be in touch regarding any additional information that may be required.

Please note that *Nature Computational Science* is a Transformative Journal (TJ). Authors may publish their research with us through the traditional subscription access route or make their paper immediately open access through payment of an article-processing charge (APC). Authors will not be required to make a final decision about access to their article until it has been accepted. [Find out more about Transformative Journals](https://www.springernature.com/gp/open-research/transformative-journals)

If you have any questions about our publishing options, costs, Open Access requirements,

or our legal forms, please contact ASJournals@springernature.com

Acceptance of your manuscript is conditional on all authors' agreement with our publication policies (see <https://www.nature.com/natcomputsci/for-authors>). In particular your manuscript must not be published elsewhere and there must be no announcement of the work to any media outlet until the publication date (the day on which it is uploaded onto our web site).

Before your manuscript is typeset, we will edit the text to ensure it is intelligible to our wide readership and conforms to house style. We look particularly carefully at the titles of all papers to ensure that they are relatively brief and understandable.

Once your manuscript is typeset, you will receive a link to your electronic proof via email with a request to make any corrections within 48 hours. If, when you receive your proof, you cannot meet this deadline, please inform us at rjsproduction@springernature.com immediately.

If you have queries at any point during the production process then please contact the production team at rjsproduction@springernature.com. Once your paper has been scheduled for online publication, the Nature press office will be in touch to confirm the details.

Content is published online weekly on Mondays and Thursdays, and the embargo is set at 16:00 London time (GMT)/11:00 am US Eastern time (EST) on the day of publication. If you need to know the exact publication date or when the news embargo will be lifted, please contact our press office after you have submitted your proof corrections. Now is the time to inform your Public Relations or Press Office about your paper, as they might be interested in promoting its publication. This will allow them time to prepare an accurate and satisfactory press release. Include your manuscript tracking number NATCOMPUTSCI-23-0725B and the name of the journal, which they will need when they contact our office.

About one week before your paper is published online, we shall be distributing a press release to news organizations worldwide, which may include details of your work. We are happy for your institution or funding agency to prepare its own press release, but it must mention the embargo date and Nature Computational Science. Our Press Office will contact you closer to the time of publication, but if you or your Press Office have any inquiries in the meantime, please contact press@nature.com.

We welcome the submission of potential cover material (including a short caption of around 40 words) related to your manuscript; suggestions should be sent to Nature Computational Science as electronic files (the image should be 300 dpi at 210 x 297 mm in either TIFF or JPEG format). We also welcome suggestions for the Hero Image, which appears at the top of our [home page](http://www.nature.com/natcomputsci); these should be 72 dpi at 1400 x 400 pixels in JPEG format. Please note that such pictures should be selected more for their aesthetic appeal than for their scientific content, and that colour

images work better than black and white or grayscale images. Please do not try to design a cover with the Nature Computational Science logo etc., and please do not submit composites of images related to your work. I am sure you will understand that we cannot make any promise as to whether any of your suggestions might be selected for the cover of the journal.

Best regards,

Ananya Rastogi, PhD
Senior Editor
Nature Computational Science

P.S. Click on the following link if you would like to recommend Nature Computational Science to your librarian: <https://www.springernature.com/gp/librarians/recommend-to-your-library>

** Visit the Springer Nature Editorial and Publishing website at <http://editorial-jobs.springernature.com> for more information about our career opportunities. If you have any questions please click [here](mailto:editorial.publishing.jobs@springernature.com).**